# Gene-diet interactions associated with complex trait variation in an advanced intercross outbred mouse line

Artem Vorobyev[1,2,15], Yask Gupta[1,15], Tanya Sezin[2,15], Hiroshi Koga [1,12], Yannic C. Bartsch[3], Meriem Belheouane[4,5], Sven Künzel[4], Christian Sina[6], Paul Schilf[1], Heiko Körber-Ahrens[1,13], Foteini Beltsiou [1], Anna Lara Ernst [1], Stanislav Khil'chenko [1], Hassanin Al-Aasam [1], Rudolf A. Manz [7], Sandra Diehl[8], Moritz Steinhaus[3], Joanna Jascholt[1], Phillip Kouki[1], Wolf-Henning Boehncke[9], Tanya N. Mayadas[10], Detlef Zillikens [2], Christian D. Sadik [2], Hiroshi Nishi[10,14], Marc Ehlers [3], Steffen Möller [11], Katja Bieber [1], John F. Baines[4,5], Saleh M. Ibrahim[1] & Ralf J. Ludwig [1]

Phenotypic variation of quantitative traits is orchestrated by a complex interplay between the environment (e.g. diet) and genetics. However, the impact of gene-environment interactions on phenotypic traits mostly remains elusive. To address this, we feed 1154 mice of an autoimmunity-prone intercross line (AIL) three different diets. We find that diet substantially contributes to the variability of complex traits and unmasks additional genetic susceptibility quantitative trait loci (QTL). By performing whole-genome sequencing of the AIL founder strains, we resolve these QTLs to few or single candidate genes. To address whether diet can also modulate genetic predisposition towards a given trait, we set NZM2410/J mice on similar dietary regimens as AIL mice. Our data suggest that diet modifies genetic susceptibility to lupus and shifts intestinal bacterial and fungal community composition, which precedes clinical disease manifestation. Collectively, our study underlines the importance of including environmental factors in genetic association studies.

[1] Lübeck Institute of Experimental Dermatology and Center for Research on Inflammation of the Skin, University of Lübeck, Ratzeburger Allee 160, 23562 Lübeck, Germany. [2] Department of Dermatology and Center for Research on Inflammation of the Skin, University of Lübeck, Ratzeburger Allee 160, 23562 Lübeck, Germany. [3] Laboratories of Immunology and Antibody Glycan Analysis, Institute for Nutritional Medicine, University of Lübeck and University Medical Center Schleswig-Holstein, Ratzeburger Allee 160, 23562 Lübeck, Germany. [4] Max Planck Institute for Evolutionary Biology, August-Thienemann-Straße 2, 24306 Plön, Germany. [5] Institute for Experimental Medicine, Kiel University, Christian-Albrechts-Platz 4, 24118 Kiel, Germany. [6] Institute of Nutritional Medicine, Molecular Gastroenterology, University of Lübeck, Ratzeburger Allee 160, 23562 Lübeck, Germany. [7] Institute for Systemic Inflammation Research, University of Lübeck, Ratzeburger Allee 160, 23562 Lübeck, Germany. [8] Department of Dermatology, Venereology and Allergology, Goethe University, Theodor-Stern-Kai 7, 60590 Frankfurt am Main, Germany. [9] Divison of Dermatology and Venereology, Geneva University Hospitals, and Department of Pathology and Immunology, University of Geneva, Rue Gabrielle-Perret-Gentil 4, 1205 Genève, Switzerland. [10] Center for Excellence in Vascular Biology, Department of Pathology, Brigham and Women's Hospital and Harvard Medical School, 75 Francis St, Boston, MA 02115, USA. [11] Institute for Biostatistics and Informatics in Medicine and Ageing Research, Ernst-Heydemann-Str. 8, 18057 Rostock University of Rostock, Germany. [12] Present address: Department of Dermatology, Kurume University School of Medicine, 67 Asahimachi, Kurume, Fukuoka 830-0011, Japan. [13] Present address: Department of Urology, University Medical Center Goettingen, Robert-Koch-Strasse 40, 37075 Goettingen, Germany. [14] Present address: Department of Nephrology and Endocrinology, University of Tokyo, 7 Chome-3-1 Hongo, Bunkyo City, Tokyo 113-8654, Japan. [15] These authors contributed equally: Artem Vorobyev, Yask Gupta, Tanya Sezin. Correspondence and requests for materials should be addressed to R.J.L. (email: ralf.ludwig@uksh.de)

In humans, genome-wide association studies (GWAS) have identified hundreds of genetic variants associated with complex human diseases and traits, providing detailed insights into their genetic architecture[1]. However, depending on the phenotypic trait, only 5–50% of the variation is explained by host genetics while rest remains unexplained[2,3]. Recently, attention has shifted on the environment and its interaction with host genetics as a key regulator of complex traits[4]. Gene-by-environment interactions (GxEs) occur when environmental factors and genetic variation have a joint impact on disease susceptibility, thus deconstructing their individual contributions[4]. These interactions are thought to explain a large proportion of the unexplained variance in heritability[5]. For instance, the interaction of genetics (e.g., the HLA locus) with environment (e.g., smoking) exemplifies the joint genetic and environmental control of the risk of developing rheumatoid arthritis (RA). Thus, while both presence of the *HLA-DRB1* haplotype and smoking confer a similar risk of developing RA, the risk increases fourfold if both factors are present[6]. Furthermore, dietary or microbe-derived metabolites can induce inflammation by modulating specific receptor responses in the gut, further suggesting that the environment contributes to complex physiological traits[7].

With diet being a major constituent of an organism's environment, we hypothesized that diet alone and its interaction with host genetics may account for a considerable proportion of phenotypic variability in complex traits. Our interest in this topic was further provoked by the clinical observation of metabolic and cardiovascular comorbidity in chronic inflammatory diseases[8]. One school of thought considers inflammation a key driver of metabolic and cardiovascular comorbidity, while the other suggests that this comorbidity may be a result of a joint genetic control. Meta-analysis of GWAS data, however, has documented little overlap of risk alleles among inflammatory, metabolic, and cardiovascular diseases[9]. In contrast, increased food intake has been suggested as a more probable risk factor for developing these diseases[10]. Nevertheless, little experimental evidence exists in favor of either hypothesis. To address this controversy and, unravel the impact of diet on complex traits, we expose a large colony of an advanced intercross outbred mouse line (AIL) to three different diets: caloric restriction, Western diet, and control diet. The overall experimental rationale is to mimic dietary lifestyles in their extremes, such as normal control diet, Western diet mimicking the food of the modern Western countries, as well as deficit of food intake in developing countries. A total of 1154 mice are genotyped and phenotyped for 55 physiological and pathophysiological traits.

Our results suggest that, for many traits, diet in addition to genotype or gene-diet interaction, explains a large portion of the phenotypic variation. Based on publicly available and herein generated genome sequence data of the parental mouse strains of the AIL mice, we fine-map several of the quantitative trait loci (QTLs) to variants in few or even single genes. Most importantly, the landscape of genomic association of traits changes considerably when diet is considered as an interactive variable with host genome. To address whether diet-modulated genetic association is functionally relevant, we select one parental strain of the AIL mice, the NZM2410/J, as it was most susceptible to gene-diet interaction in our study. This strain develops spontaneous pathological phenotypes, such as antinuclear antibodies (ANA) and lupus. We expose NZM2410/J mice to the same three diets. Under caloric restriction, all NZM2410/J mice are protected from lupus development and only 5% produce ANA, whereas animals fed a Western diet succumb due to an accelerated and severe lupus and more than 50% of the mice produce ANA. To better understand the underlying mechanisms of diet-modulated lupus development in NZM2410/J mice, we perform longitudinal

analysis of their intestinal micro- and mycobiota, as well as RNA-sequencing (RNA-Seq) of their spleens. This reveals that diet-induced changes in the intestinal micro- and mycobiota precede clinical disease manifestation in NZM2410/J mice, and are associated with ANA production. Furthermore, by associating diet and pathophysiological traits (e.g., lupus and ANA) with the RNA-Seq data, we identify dysregulated genes and biological pathways that predispose NZM2410/J mice to disease onset and production of ANA. Finally, we use our multi-omics data to fine-map QTL for ANA production in AIL mice.

## Results

**Impact of diet and host genetics on complex traits**. A large cohort of male and female mice from an autoimmunity-prone AIL was fed three different diets (caloric restriction, control-, and Western-diet) until an age of 24 weeks. Thereafter, mice were genotyped and phenotyped. In total, we quantified 55 phenotypes that were defined as either physiological or pathophysiological. Phenotypes and assessment methods are listed in Supplementary Table 1. Physiological phenotypes were further categorized into metabolic, hematological, immunoglobulin, glycosylation pattern, and other phenotypes. Ultimately, the effects of host genetics, diet, and sex on the phenotypic variation of each trait were analyzed as stated in "Methods" (Fig. 1a, b). Diet accounted for the largest proportion of the phenotypic variation in metabolic (for example, 48% of the phenotypic variability for final body weight is explained by diet), immunoglobulin (up to 37% for IgA/IgM ratio), and pathophysiological traits (up to 46% NAFLD ballooning; Supplementary Table 1, Supplementary Data 1).

In contrast, diet had little impact on phenotypic variability of differential blood counts (up to 6% for eosinophils). For IgG glycosylation traits, specifically composition of the biantennary sugar residue at Asn297, comparable contributions were observed for both host genetics (up to 11% for the ratio between sialic acid and galactose) and diet (up to 12.5% for G1). Other known covariates, such as sex, only explain a small magnitude of the phenotypic variation in the studied traits (up to 8% for body weight at 2 months of age), Supplementary Table 1, Supplementary Data 1.

Next, to more deeply characterize the impact of genetics, and identify genomic loci that co-vary with the investigated phenotypes, we performed QTL mapping. At the genome-wide level ($\alpha_{gw} < 0.05$), we identified 21 QTL, corresponding to 18 phenotypes (Fig. 1b, Supplementary Data 1). In addition, 20 suggestive (chromosome-wide) QTL for 13 distinct phenotypes (Supplementary Data 2) were found (Fig. 1b). Overall, host genetics accounted for 2–28% of the phenotypic variation, with a strong genetic association for a cis-QTL for C-reactive protein (CRP) (LOD = 36.5), and a previously reported QTL for coat color on chromosomes 2 (LOD = 28.4) and 7 (LOD = 29.2)[11] (Supplementary Data 1).

**Diet reveals hidden genetic associations**. In addition to QTL exclusively associated with host genetics, we next investigated the impact of the interaction between host genetics and diet in the measured phenotypes. Here, we identified 23 gene-diet-associated QTL for a total of 11 phenotypes at the genome-wide significance level (Fig. 1b, Supplementary Data 1). At the chromosome-wide significance level, 114 additional, suggestive QTL correlated with a total of 38 phenotypes (Supplementary Data 2). Considering diet as an interacting variable in the QTL mapping led to a shift of the genetic association that has been mapped when solely focusing on genetics. For instance, while host genetics is associated with body weight only at early time points (2 months), additional QTL are observed at 2 months as well as at later time

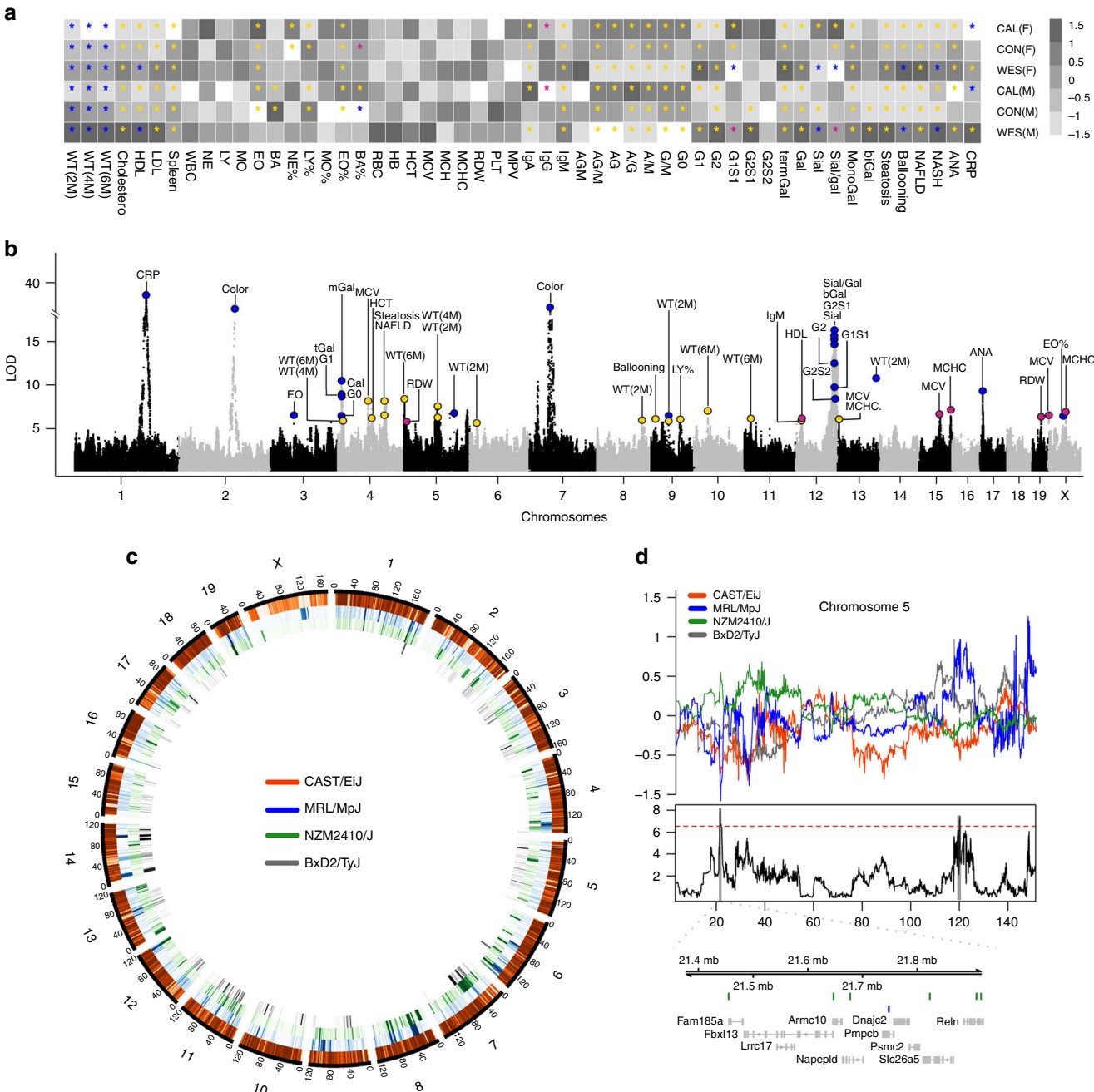

**Fig. 1** Phenotyping, QTL identification, and fine mapping of dietary perturbations in an AIL mouse population. **a** Heatmap illustrating distribution (normalized between 1.5 to −1.5) of the investigated complex traits in AIL mice for the three dietary regimens in both sexes. Data were analyzed using Kruskal–Wallis test followed by Dunn's test for multiple comparisons (P-value was adjusted by Sidak correction procedure). For binary phenotypes (ANA and NASH), chi-square test was performed followed by Fisher exact test for multiple comparisons where p-values were adjusted by Benjamini-Hochberg correction procedure. The asterisk (*) within each box indicates statistical significance (Padj < 0.05), whereas the different color codes indicate the association of the particular trait with either diet (yellow), sex (pink) or both (blue). Extension of the abbreviations of the traits and their sample size (n) are provided in Supplementary table 1. **b** Manhattan plot showing QTL ($\alpha_{gw}$ < 0.05) in the AIL population associated with various complex traits. The x-axis describes chromosomes in the C57BL/6J mouse genome (reference mouse assembly mm10), while the y-axis represents the LOD score (log of odds ratios). The QTL marked with blue represent the additive model, while gold and pink represent diet- and sex-interacting QTL, respectively. **c** An overview of variants called from 4 founder mouse strains (BxD2/TyJ, MRL/MpJ, NZM2410J, and CAST/EiJ) compared to the C57BL/6J mouse genome reference assembly (mm10). Each track in the circle indicates the relative density of single nucleotide polymorphisms (SNPs) and insertions and deletions (indels) for four strains across mouse chromosomes. **d** Schematic representation of fine-mapped genes for weight (6 months) under the influence of diet using whole-genome sequencing. The uppermost panel shows founders coefficients for significant QTL in chromosome 5. The middle panel shows a line plot for QTL with a red line indicating a genome-wide significance level ($\alpha_{gw}$ < 0.05). The lowermost panel shows the genes present within the QTL and variants differing between NZM2410/J and MRL/MpJ strains, leading to the exclusion of several candidate genes. Source data for (**a**) are provided in the Source Data file

points (4 and 6 months) when considering diet as an interacting variable (Supplementary Data 1). Moreover, gene-diet-associated QTL, derive mainly from the genome of the NZM2410/J and the MRL/MpJ mouse, were also identified for the pathophysiological traits such as NAFLD (LOD = 6.3), steatosis (LOD = 7.9), and ballooning (LOD = 5.9). Taken together, these results show that diet changes the genetic association of several of the investigated complex traits. To understand the impact of the sex on complex traits, we next considered sex as an interacting variable in our QTL-mapping study. The gene/sex-co-regulated QTL accounted mainly for hematological and metabolic parameters. In detail, four phenotypes were associated with 7 QTL at genome-wide significance, and an additional 30 phenotypes show an association with 88 QTL at the chromosome-wide significance level (Fig. 1b, Supplementary Data 1–2, URL: http://diet.ag-ludwig.com). The impact of sex on complex traits points towards an interaction of autosomal with idiochromosomal genes.

**QTL fine-mapping using whole-genome sequencing**. One of the main challenges in QTL-mapping studies is to reach sufficient resolution to identify few- to single candidate genes. In our cross, the average size of the QTL we defined at genome-wide- significance level ($\alpha_{gw} < 0.05$) is 1.57 ± (SEM) 1.59 Mb. Therefore, to further enhance the QTL resolution, we sequenced the genomes (whole-genome sequencing, ~30× coverage) of three parental strains, i.e., BxD2/TyJ, MRL/MpJ, and NZM2410/J and derived the genome of CAST/EiJ mice from a publicly available database[12]. Detailed assembly statistics for the four strains is mentioned in Supplementary Data 3. Since these strains are inbred, we exclusively called homozygous single nucleotide polymorphisms (SNPs), insertions and deletions (indels). In comparison to the C57B6/J (mm10) genome, we found 5,203,605 SNPs and 1,054,204 indels in the MRL/MpJ strain, 5,612,844 SNPs and 1,111,081 indels in the NZM2410/J strain, and 2,562,122 SNPs and 513,925 indels in the BxD2/TyJ strain. In comparison to other sequenced strains (Mouse Genomes Project), 1.5%, 1.8%, and 2.9% were novel SNPs, and 29.3%, 32.9%, and 33.9% were novel indels for BxD2/TyJ, MRL/MpJ, and NZM2410/J mice strains, respectively[13] (Fig. 1c, Supplementary Data 3). Comparison of the genomic landscape of founder strains (Fig. 1d) reduced the number of candidate genes for the 51 identified QTL ($\alpha_{gw} < 0.05$), corresponding to 30 phenotypes (Supplementary Data 1).

Next, we categorized QTL defined at genome-wide significance into three groups according to the degree of evidence supporting the candidacy of the gene for the respective trait: The first group comprises genes whose candidacy is either supported by human GWAS, a corresponding spontaneous phenotype in knockout mice or a presence of a cis-QTL (Supplementary Data 1). This group of genes contained 58% of the overall identified genes in our study. Of these, 68% were controlled by gene-diet interaction, followed by genes exclusively controlled by host genetics (18%) and gene-sex interaction (14%). For example, two of the host genetics-associated genes, *B4galt1* and *Igh*, were fine mapped to loci on chromosomes 4 and 12, respectively and have been shown to contribute to glycosylation modifications in humans[14]. In detail, deficiency in *B4galt1* leads to a severe congenital neurological disorder of glycosylation type IId. The *Igh* locus was recently discovered to be associated with IgG N-glycosylation in human GWAS[15]. Among the genotype-by diet-associated candidate genes, a locus on chromosome 5 containing the *Napepld* gene contributed to body weight at age 4 and 6 months. *Napepld* encodes for N-acyl phosphatidylethanolamine phospholipase D, which has a known function in hydrolyzing diet-induced N-acylphosphatidylethanolamines[16]. Deletion of

*Napepld* in adipocytes leads to obesity, glucose intolerance and adipose inflammation[17]. Similarly, among the genes associated with gene-diet interaction, we identified the *Nr4a3* gene at a locus for body weight, which was suggested as a potential target for amelioration of insulin resistance as well as treatment of type 2 diabetes and metabolic syndrome[18].

The second category of identified QTL comprised eight loci with genes whose candidacy is strongly corroborated by published evidence. As an example, among the genes regulated solely by host genetics, *Cux1* and *Cysltr1* were associated with body weight and eosinophil abundance, respectively. *Cux1* is known to modulate food intake in both rodents and humans, and *Cysltr1* governs eosinophil influx and migration into tissues[19]. Furthermore, within the group of loci associated with the gene-diet interaction, *Nfkbia* and *Casp9* were present. The *Nfkbia* is associated with immunoglobulin M (IgM) concentrations, and mice harboring a germline mutation in *Nfkbia* exhibit defective B cell maturation and antibody production[20]. Both steatosis and NAFLD were associated with *Casp9*, and described as a potential marker for hepatocyte apoptosis during the development of NAFLD[21].

The remaining eight QTL, comprising the third group, contain candidate genes that were yet to be associated for their biological relevance with the observed phenotypes. The majority of the candidate genes in this group were associated with physiological phenotypes such as body weight (*Parp8*, *Ube2cbp*, *Sec61b*, and *Ubl3*) and hematological parameters (*Kcnd3*, *Diaph2*, and *Sorc3*).

**Diet modulates genetic susceptibility**. To this end, we demonstrate that diet shifts the genetic association and uncovered multiple genes associated with metabolic and pathophysiological traits. Nonetheless, it was unclear whether this diet-mediated effect of genetic association is of functional relevance. While addressing gene-diet interactions models, we observed that the NZM2410/J and the MRL/MpJ strains were the major contributors to phenotypic variation in the AIL population (Supplementary Data 1, URL: http://diet.ag-ludwig.com). Both strains are prone to the spontaneous development of autoimmune diseases, such as lupus and pancreatitis[22,23]. Independently, we also observe a host genetics-associated locus for ANA, a characteristic feature of lupus, mapped to the genome of the NZM2410/J mice. To address whether diet has a functional impact on genetic disease susceptibility, we exposed NZM2410/J mice to the same three dietary regimens as for the AIL population. Subsequently, the prevalence of ANA and lupus development were monitored over time. As expected, mice on caloric restriction gain less weight over time than mice on control or Western diet (Fig. 2a, b). Regarding lupus development, 41% of mice on control diet develop lupus nephritis. In contrast, none of the mice on caloric restriction and almost all mice (90%) on Western diet develop lupus. Of note, in the Western diet group, disease onset, as measured by proteinuria, was accelerated, with an average onset on week 20.0 ± (SEM) 3.0 compared to week 24.0 ± (SEM) 2.7 in the control group (Fig. 2c). The clinical findings were further supported by histopathological changes in the kidneys. Specifically, crescent formation and frequency of periodic acid–Schiff (PAS) positive deposits were lowest in mice held at caloric restriction and highest in mice fed Western diet (Fig. 2d–f). In line with the clinical and histological observations, only 5.6% of mice on caloric restriction had circulating ANA, while this increased to 52.6% in mice on Western diet (Fig. 2g). Collectively, our results indicate that diet modulates the genetic susceptibility to develop lupus. Next, to delineate changes induced by diet that resulted in differential susceptibility to lupus, we studied gut flora (longitudinally) and transcriptomic alterations in these mice.

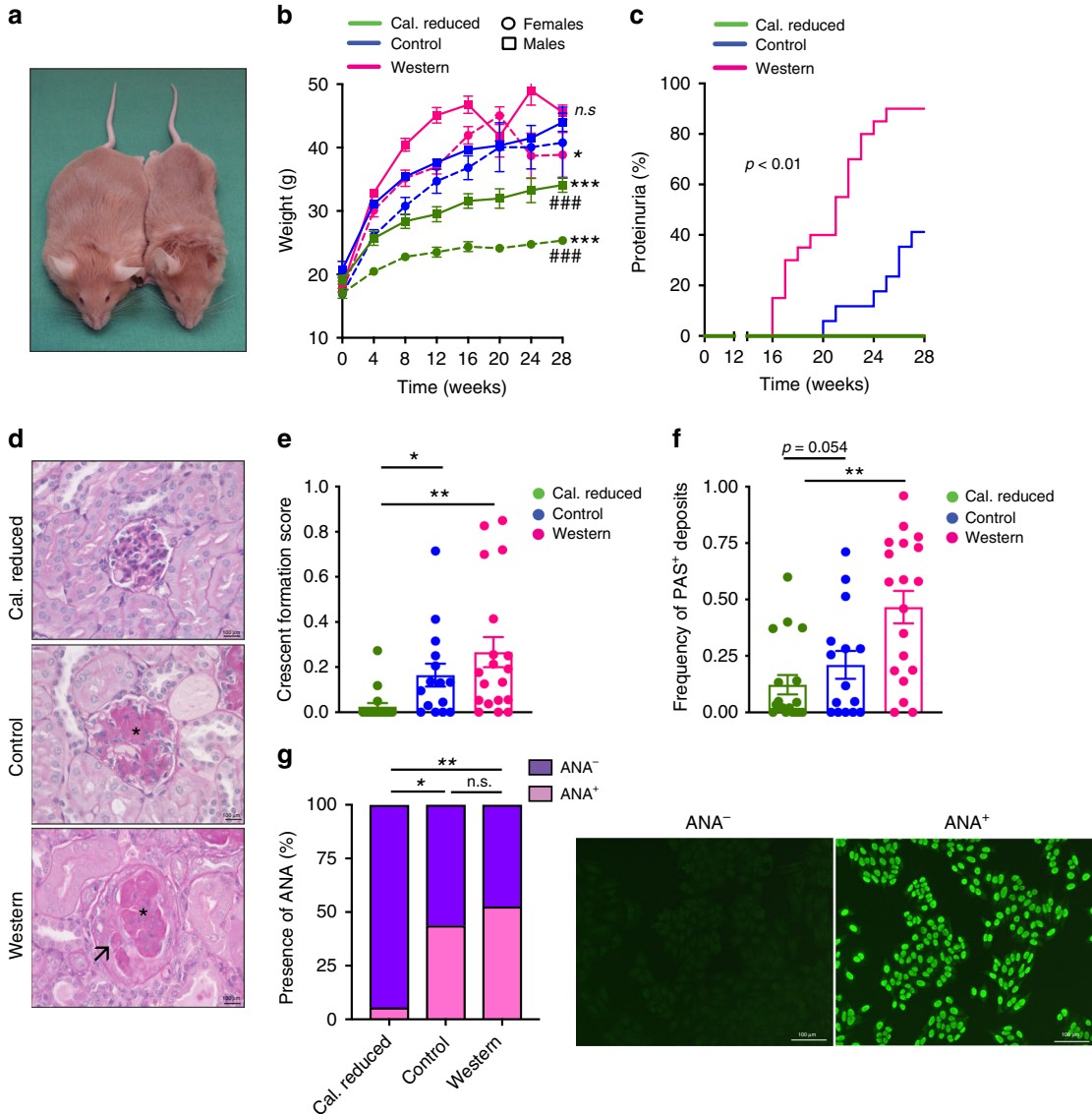

**Fig. 2** Impact of diet on development of lupus nephritis in NZM2410/J mice. **a** Representative image of NZM2410/J mouse exposed to Western diet (left) vs. a mouse fed with calorie-restricted diet (right) after 28 weeks of dietary intervention. **b** X-Y plot demonstrating the impact of diet and sex on body weight for caloric restriction ($n_{females} = 8$, $n_{males} = 10$), for control diet ($n_{females} = 7$, $n_{males} = 10$) and for western diet ($n_{females} = 9$, $n_{males} = 11$). Error bars indicate means ± SEM. Data was analyzed using two-way ANOVA followed by Tukey's multiple comparison test, *$p < 0.05$, ***$p < 0.001$, n.s., not significant. All were evaluated in comparison to mice on control diet. ###$p < 0.001$ compared with mice on Western diet. **c** Step plot showing the onset of proteinuria in NZM2410/J mice for caloric restriction ($n = 18$), control ($n = 17$) and western ($n = 20$) diets. Mice fed Western diet developed proteinuria four weeks earlier (20.0 ± 3.0 weeks) in comparison to mice fed control diet (24.0 ± 2.7 weeks). Notably, none of the mice on caloric restriction demonstrated signs of proteinuria throughout the entire observation period. Data are shown as the mean ± SEM and were analyzed using two-way ANOVA. **d** Representative histological images of glomerular changes in NZM2410/J mice show that mice on caloric restriction demonstrated almost no signs of glomerular damage. In contrast, mice on the Western diet had severe histological changes, such as crescent formation (arrow) and PAS-positive deposits in glomeruli (asterisk), which were milder in the control group. Scale bar, 100 μm. Bar plots demonstrating crescent formation (**e**) and sclerotic changes (**f**) in NZM2410/J mice show significantly more severe kidney injury in Western diet mice compared to mice on caloric restriction (caloric restriction, $n = 18$; control diet, $n = 15$; Western diet, $n = 19$). Error bars indicate means ± SEM. Data was tested for statistical significance using the Kruskal–Wallis test, with Dunn's post hoc test, *$p < 0.05$, **$p < 0.01$. **g** Quantification of antinuclear antibodies (ANA) in sera of NZM2410/J mice set on different diets as determined by indirect immunofluorescence analysis on HEp-20–10 cells (left panel; caloric restriction, $n = 18$; control diet, $n = 16$; western diet $n = 19$). Data was analyzed using the X2 test with Fisher exact test as a post hoc test. *$p < 0.05$, **$p < 0.01$, n.s., not significant. Representative images of ANA negative and positive sera samples (right panel). Scale bar, 100 μm. Source data for (**b**, **c**, **e**–**g**) are provided in the Source Data file

**Diet-associated changes of the microbiota before disease onset.** Previous studies documented that diet changes the composition of the intestinal bacterial and fungal communities[24,25]. Furthermore, alterations in the gut micro- and mycobiome are frequently associated with disease phenotypes, including SLE[26,27]. However, the majority of these studies present cross-sectional data collected following manifestation of disease phenotype. Thus, it remains largely unknown whether the observed alterations in the microbiota composition occur prior to disease or as a consequence of the disease phenotype.

Therefore, we performed longitudinal sampling of feces from lupus-prone NZM2410/J mice that were set on the same diet as

the AIL mice. We categorized samples into three stages, i.e., (1) collected immediately after weaning (naïve); (2) at a transient state in which all mice were phenotypically healthy (i.e., absence of proteinuria); and (3) at a final time point, which was either at the end of the observation period or at the time when mice had to be sacrificed because of severe, lupus-nephritis-induced weight loss. Afterwards, using amplicon- based next generation sequencing, samples were investigated for microbial (V1–V2) and mycobial (ITS2) composition. For microbiota, species diversity (alpha diversity, Chao1 index) among the different diets was significant in the naïve, transient and final stages, showing a higher diversity in mice on calorie-reduced diet (Fig. 3a). Most likely, the identified differences in the microbial composition at this early stage reflect the immature gut microbiome present at this age[28]. Differences in alpha diversity were also found for transient and final stages when stratifying the mice for presence or absence of lupus (Fig. 3a). Notably, at the transient stage, none of the mice show any signs of disease, but the samples were characterized based on the establishment of future disease at the final time point. Therefore, our results indicate that changes in microbial communities occur before the actual clinical disease manifestation. With respect to the mycobiome, we did not find differences in fungal richness (Chao1 index) at any stage for the stratified subgroups (diet and disease; Supplementary Fig. 1a). However, the observed species diversity (Shannon index) showed differences between diseased and non-diseased mice at the transient stage (Fig. 3b).

For beta-diversity (UniFrac distances), we observe significant changes in the microbial composition across different diets and disease at the transient and final stages (Fig. 3c; Supplementary Fig. 1b). For the mycobiota, differences in the beta-diversity (Jaccard distances) were found when stratifying the mice based on diet at transient and final stages (Fig. 3d; Supplementary Fig. 1c). To identify, potential microbial and fungal biomarkers for traits (diet and disease), we used the LEfSe algorithm[29]. The linear discriminant analysis effect size (LEfSe) method combines standard statistical tests with biological consistency and effect relevance to determine the features (taxonomical ranks) that most likely explain the differences between classes (such as diet and disease). For mycobiome, at the transient stage, fungi such as Sporobolomyces were especially abundant in the calorie-restricted (Supplementary Fig. 2a) and non-diseased (Supplementary Fig. 2c) groups, while Wallemia were more abundant in Western diet and diseased mice (Fig. 3e). With the progression of disease, at the final stage, Cytospora were more abundant among the calorie-restricted (Supplementary Fig. 2b) and non-diseased mice (Supplementary Fig. 2d), while Capronia pilosella, Exophiala and Helotiales were abundant in the Western-diet and diseased groups. For microbiota, at transient stage Paraburkholderia, Betaproteobacteria, Dubosiella and Faecalibaculum were abundant in the calorie-restricted and non-diseased groups at both transient and final stages (Fig. 3f). In contrast, Dorea longicatena, Lachnoclostridium and Roseburia were abundant in the Western diet and disease group of mice (Supplementary Fig. 2 e–h).

**Spleen transcriptomic changes associate with diet or disease**. As germinal center formation in the spleen is a key event in autoimmune-prone mice[30], and its formation at least partially drives lupus pathogenesis[31] we next investigated splenic transcriptome in the NZM2410/J mice. In total, we identified 1415 differentially expressed genes ($P$adj < 0.05) when stratifying the data based on diet (Supplementary Data 4) and 1,349 differentially expressed genes when differentiating between healthy and diseased mice (Supplementary Data 5). Of these differentially expressed genes, 830 genes were common to both diet and disease

status, suggesting that this gene-set is important for disease and is modulated by diet (Supplementary Data 5). Pathways enriched in diseased mice and mice fed Western diet included the complement cascade, FCERI signaling, cytokine signaling, and neutrophil degranulation. The butyrophilin family interaction was enriched in calorie-restricted and non-diseased mice (Supplementary Data 6, Supplementary Table 2).

Given the significant changes in the gut micro- and mycobiota composition in the NZM2410/J under different diets leading to lupus, we next examined the gene signatures associated with micro- and mycobiota. In this context, we used an approach proposed by Tong and colleagues, where they show that microbiota can be clustered into functional microbial communities (FMCs) based on taxa co-occurrences patterns[32]. To determine such ecological structures, they first construct microbial co-occurrence networks. Nodes of these networks, representing OTUs were grouped based on their topological overlaps using hierarchical clustering and were termed as FMCs. Such an approach provides dimensionality reduction (eigenOTUs; that can be described as first principle component of FMCs to summarize the OTU abundances in a given community), which can be used to associate communities with multiple traits. We applied this strategy on the gut microbial and fungal communities. First, for the microbiome data, we find that out of the six clustered modules, FMC1, comprised of 105 OTUs, was significantly upregulated in diseased and Western diet fed mice (Fig. 4a). Similarly, we identified classified functional fungal communities (FFCs) and correlated them with sex, disease state, stages of sampling, and diet. We identified 6 FFCs, of which FFC4, consisting of 24 OTUs, was associated with diet and disease progression (Fig. 4b). It was suggested that fungi and their metabolites can affect targeted bacterial species[33]. To explore potential correlations between the FMC1 and FFC4 we used the SparCC algorithm[34] (Fig. 4c, Supplementary Fig. 3). While the majority of taxa showed negative inter-domain correlations, such as Eisenbergiella tayi and Chytridiomycota, other taxa, including Ruminococcus torques and Pezizaceae, showed the strongest positive inter-domain correlation (Fig. 4c). This suggests that diet may modulate complex intestinal micro-ecosystems that contribute to disease pathogenesis.

Next, to explore potential host disease-associated molecular mechanisms modulated by micro- and mycobiota we correlated the eigenOTU values of FMC1 and FFC4 to the expression levels of genes derived from NZM2410/J mice. We identified 679 and 116 differentially expressed genes associated with FMC1 (Supplementary Data 7) and FFC4, respectively (Supplementary Data 8). Of these genes, 43 genes were common between micro- and mycobiota, and 38 genes contributed to disease and were also dictated by diet. The 38 genes were enriched for ontology terms classified as biological processes ($P$adj < 0.01), specifically defense response to virus, regulation of innate immune response, and regulation of ribonuclease activity (Fig. 4d).

**Candidate gene associated with ANA**. In AIL mice, ANA prevalence was lowest in mice on caloric restriction compared to mice set on control or Western dietary regimens (Fig. 5a). Moreover, ANA prevalence was associated with host genetics (Fig. 1b). Specifically, a non-diet-interactive QTL for ANA mapped to the MHC locus on chromosome 17 (Fig. 5b). For the peak SNP (UNC27797171) within this QTL, the allele derived from NZM2410/J accounted for ANA prevalence (Fig. 5c). By examining the NZM2410/J specific consequential mutation derived from whole-genome sequencing, we further shortlisted the set of 81 genes within the ANA-associated locus to 58 candidate genes. To further fine-map this QTL, we used the splenic

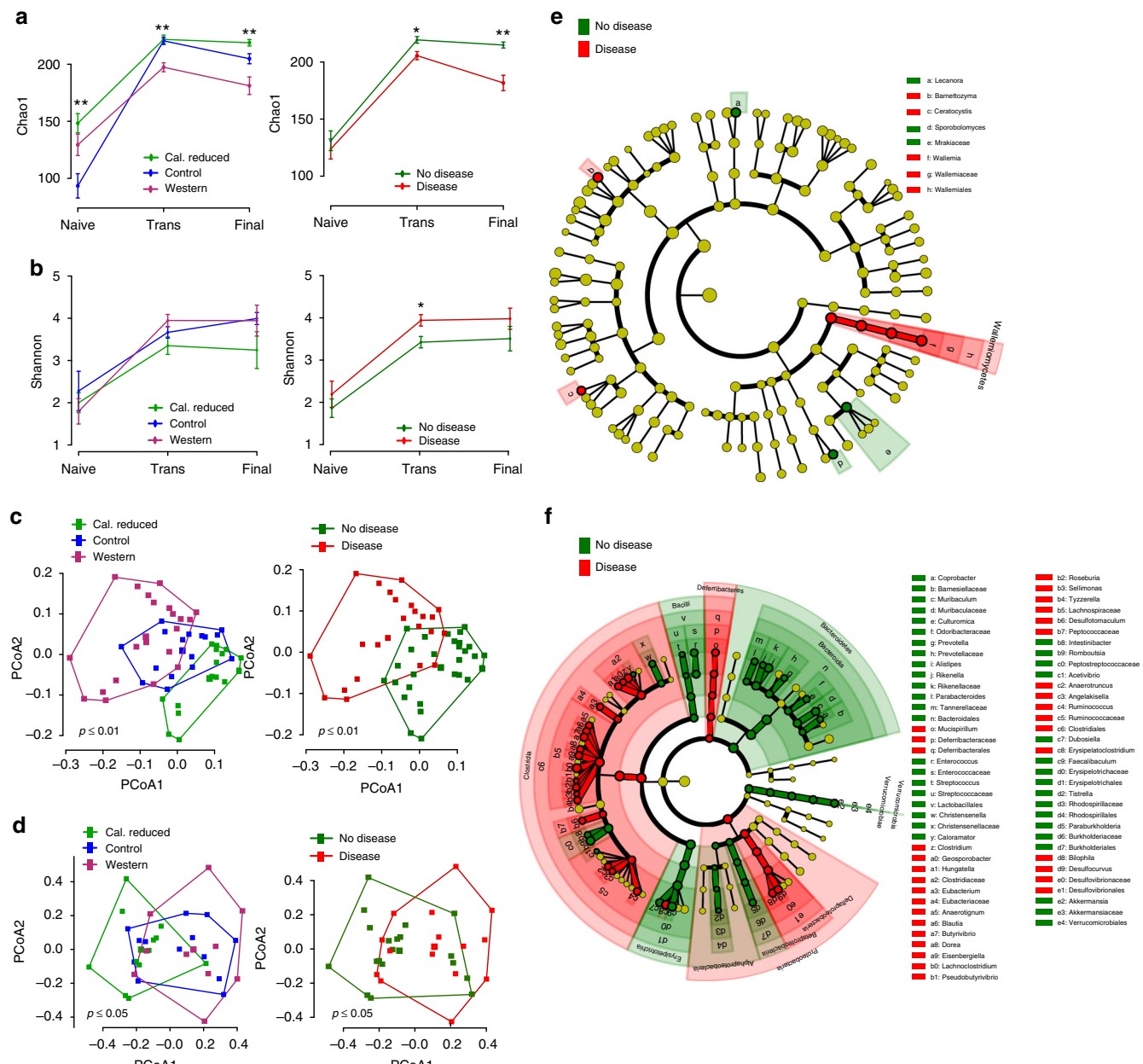

**Fig. 3** Alpha/beta diversities and most differentially abundant taxa of micro- and mycobiota for classes (diet and disease) in NZM2410/J mice. **a** Alpha diversity (Chao1 index, species richness) of microbial species between NZM2410/J mice set on three dietary regimens across the naive, transient, and final stages of disease development stratified based on caloric restriction ($n = 18$), control ($n = 16$) or western ($n = 20$) diet (left panel) or absence ($n = 31$) and presence ($n = 23$) of disease (right panel). **b** Alpha diversity (Shannon index, species richness) of the mycobiota between mice set on different diets across naïve, transient and final stages of disease stratified based on caloric restriction ($n = 12$), control ($n = 12$) and Western ($n = 10$) diets (left panel) or presence ($n = 15$) and absence ($n = 19$) of disease (right panel). **c** Principal coordinate analysis (PCoA) of beta diversity for the microbiota at the final stage of disease based on unweighted UniFrac distances between NZM210/J mice classified based on caloric restriction ($n = 18$), control ($n = 16$) or western ($n = 20$) diets (left panel) or presence ($n = 23$) or absence ($n = 31$) of disease (right panel). **d** PCoA of beta diversity for mycobiota at the final stage of disease based on Jaccard distances between NZM210/J mice classified based on caloric restriction ($n = 12$), control ($n = 12$) and western ($n = 10$) diets (left panel) or presence ($n = 15$) or absence ($n = 19$) of disease (right panel). Differentially abundant (**e**) mycobiota and (**f**) microbiota taxa identified by the LEfSe algorithm (default parameters) at transient stage in diseased vs. non-diseased mice. In panels (**a–b**), data is presented as means ± SEM and statistical significance was assessed using Kruskal–Wallis test with Mann–Whitney $U$ test as a post hoc test. The p-values were adjusted using the Benjamini-Hochberg correction procedure. *$P$adj < 0.05, **$P$adj < 0.01. Data in panels (**c–d**) were analyzed using the adonis function in R (permutations = 999). Source data for (**a–f**) are provided in the Source Data file

RNA expression data generated in NZM2410/J mice, who similarly to the AIL population harbor identical patterns of ANA prevalence among the various diets (Figs. 2g, 5a). Based on the presence or absence of ANA we observed 1,458 differentially expressed transcripts (Supplementary Data 9). Of these, 11 genes

(*H2-Eb2*, *Myo1f*, *Tap1*, *Cfb*, *C2*, *H2-K1*, *Psmb9*, *Psmb8*, *Tnxb*, *Rps28*, and *Col11a2*) were shared among the above shortlisted 58 candidate genes (Fig. 5f). We next evaluated the RNA-Seq data after stratification for diet, which is a risk factor for ANA production in our study. Here, compared to the control or Western

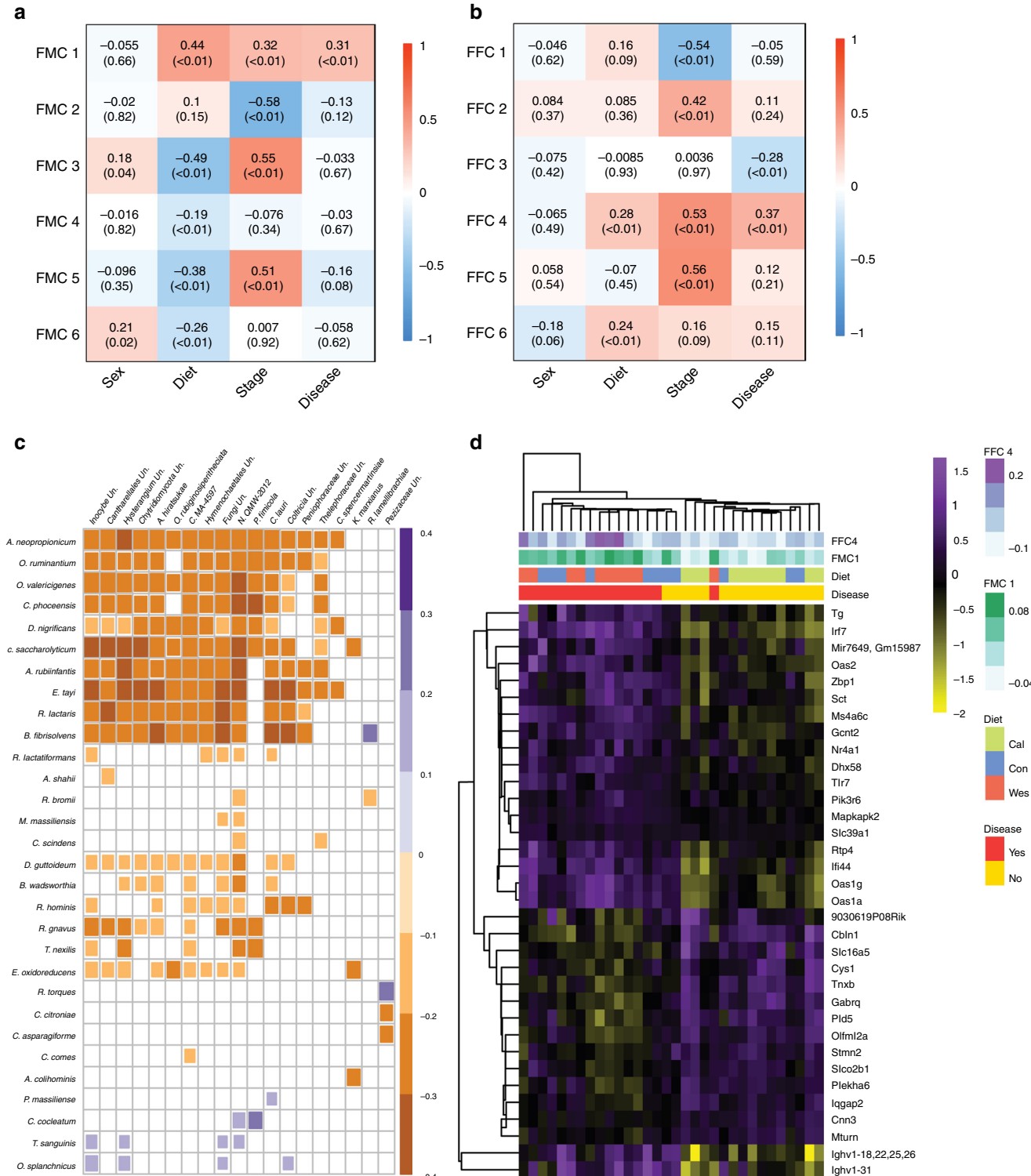

diet, 2004 genes were differentially expressed in mice held at caloric restriction (Supplementary Data 10). To further identify a diet-modulated gene controlling ANA production, we overlapped 1458 differentially expressed genes (identified for ANA presence or absence) with 2004 genes (stratification based on diet) and identified 850 shared genes between the two comparisons. Out of the 850 transcripts 8 genes overlapped with the 11 genes (*Myo1f*, *Tap1*, *Cfb*, *C2*, *H2-K1*, *Psmb9*, *Psmb8*, and *Tnxb*), which we fine-mapped for ANA QTL. Additionally, we find that ANA production in the NZM2410/J mice differed based on their intestinal

microbial and fungal composition (Fig. 5d). Out of the identified functional communities, we found FMC3, FMC5, and FFC6 to be associated with the production of ANA (Fig. 5e; Supplementary Fig. 4). After correlating the RNA-Seq data to FMC3, FMC5, and FFC6, we identified 1980, 429, and 102 differentially expressed genes, respectively (Supplementary Data 11–13). A comparison of differentially expressed genes from above analysis with fine-mapped eight genes for ANA QTL led single common candidate gene, i.e., *Tnxb* (Fig. 5g). The expression levels of Tnxb negatively correlated with both FMC3 and FFC6, and were upregulated in

**Fig. 4** Identification of functional microbial and fungal communities associated with diet and disease and their association with transcriptome data. Functional microbial (**a**) and fungal (**b**) community (FMC, FFC, respectively)–trait correlations and adjusted *P*-values. Each value within the cell without the brackets shows the Spearman correlation coefficient between covariate of interest and eigenOTUs. The adjusted *P*-values are given within the brackets in each cell were derived using Wilcoxon test for sex (microbial, $n_{females} = 24$ and $n_{males} = 31$; fungal, $n_{females} = 15$ and $n_{males} = 17$) and disease at final stage (microbial, $n_{yes} = 22/n_{no} = 31$; fungal, $n_{yes} = 14/n_{no} = 17$), and the Kruskal–Wallis test for various disease stages (microbial, $n_{naive} = 50$, $n_{trans} = 55$, and $n_{final} = 53$; fungal, $n_{naive} = 29$, $n_{trans} = 32$, and $n_{final} = 31$) and diet (microbial, $n_{calorie\ restricted} = 18$, $n_{control} = 17$, and $n_{western} = 20$; fungal, $n_{calorie\ restricted} = 11$, $n_{control} = 11$, and $n_{western} = 10$) and were further adjusted for multiple comparisons across different modules using Benjamini-Hochberg correction procedure. The degree of correlation is indicated by the color of the cell in accordance with the color legend. **c** Heat map showing the correlation between FMC1 (rows) and FFC4 (columns) OTUs (species identified by both RDP and NCBI BLAST) calculated by the SparCC algorithm. The color codes and of the cells indicate either positive (purple) or negative (orange) correlations among the species (*P*adj < 0.05), while the size of the cell positively correlates with the degree of the correlation. **d** Heatmap describing differentially expressed genes intersecting between disease ($n = 16$) and no disease ($n = 16$), diet groups i.e calorie-reduced ($n = 11$), control ($n = 11$) and western ($n = 10$) and associated with FMC1 and FFC4. The color-coded heatmap shows purple as high expression and yellow as low expression of genes (rows) among the samples belonging to different groups (columns). Source data for (**a–d**) are provided in the Source Data file

calorie-restricted mice. Collectively, our data identifies *Tnxb* as a potential critical regulator of ANA production. Furthermore, our data suggests that its expression may be modulated by calorie restriction and associated traits such as microbiota and mycobiota.

## Discussion

In this study, we show an eminent impact of diet in comparison to host genetics on both metabolic (i.e., body weight, steatosis and cholesterol levels) and immune-system-related complex traits (i.e., immunoglobulins levels) in mice. While host genetics explains (additive QTL) some of the metabolic and immune-system-related traits, its interaction with diet reveals additional associations (interactive QTL). When using sex as an interactive covariate, we identified predominantly QTL accounting for hematological parameters. Recently, a systematic review of animal research showed a vast over-representation of experiments that exclusively included mice of a single sex in their experiments. Where two sexes were included, most of the data was analyzed without taking sex into account. Using sex as a biological variable, close to 10% of categorical traits and over 50% of continuous data exhibited sexual dimorphism[35]. Herein, we show a much lesser impact of sex on the variability of complex traits. This seeming discrepancy may be best explained by the difference in mouse phenotypes investigated. To fine map the identified QTL, some of the associations were resolved to single candidate genes by utilizing the sequenced genomes of the founder strains of the AIL (NZM2410/J, BxD2/TyJ, and MRL/MpJ mice) and the publicly available genome of the CAST/EiJ mice. For instance, in addition to potential candidate genes, we also identify genes that were validated by reports in human GWAS or in vivo knockout studies (Supplementary Data 1). Next, to address if diet has a modulatory impact on host-genetics determined phenotypic traits, we fed NZM2410/J inbred mice three different diets, similar to AIL mice. NZM2410/J mice, which are genetically prone to develop lupus, also explained the majority of gene-diet-associated QTL in the AIL population. In NZM2410/J mice, caloric restriction led to a complete protection from clinical lupus manifestation. Conversely, 90% disease penetrance was observed in the mice fed a Western diet. Hence, we demonstrate that diet overrides genetic susceptibility and delays disease onset. Similar to other studies, we found that diet reshapes the gut microbiome and possibly prompts differences in disease susceptibility[36]. Additionally, we show that these effects of diet are not limited to the gut microbiome but extend to the gut mycobiome. Furthermore, our data indicates that alterations in intestinal bacterial and fungal communities precede the onset of lupus. In-depth analysis of the intestinal micro- and mycobiomes revealed that only a subset of co-occurring communities (FMC1 and FFC4) is associated with

diet over time. By exploring transcriptomics in these mice, we show that diet, micro- and mycobiome associate with immune-related genes and pathways in disease pathogenesis. We did not monitor, however, the mice for their locomotor activity, which may have been impacted by the different diets[37]. Thus, this needs to be taken into consideration as a limitation when interpreting the data of our study.

In addition, we here illustrate the strength of our collective data, and analysis strategy to fine-map complex genetic susceptibility loci. Specifically, as an example, we apply a multi-omics approach to resolve a complex additive QTL for ANA, located on the MHC locus (chromosome 17, 33–34 Mb). Though, in comparison to other mouse strains, the NZM2410/J strain, which is a cross between NZW and NZB strains, have a high ANA prevalence[38], the allele A (peak SNP UNC27797171) derived from NZM2410/J mice is associated with a low ANA prevalence. This is in accordance with previous studies, which showed that the presence of a homozygous H2 allele in the MHC locus of the NZB/BINJ mice protects from ANA production[39]. When stratified for ANA prevalence based on the observed alleles (A/G) within the peak SNP and diet, we show that mice having protecting A allele held on Western diet show high prevalence of ANA in comparison to mice on calorie-restricted diet having a susceptible allele G. Altogether, our results suggest that diet reverses the effect of host genetics in controlling a complex trait like ANA. Thus, it is tempting to speculate that under a calorie-restricted dietary regimen, one may countermand this inherited predisposition to develop disease. The shared similarity between NZM2410/J mice and AIL population in regard to ANA prevalence-dependency on diet and the microbial/mycobial composition, permitted us to integrate and examine data between the two. As a result, we used multiple datasets obtained in the course of our study to fine map a protective candidate gene, *Tnxb*, for an ANA QTL. The gene *Tnxb* has been associated with ANA-related pathophysiological phenotypes, such as lupus, in several populations[40,41]. Additionally, *Tnxb* deficiency has been reported to suppress hepatic dysfunction by suppressing inflammatory cell infiltrate induced in mice by a high-fat and high-cholesterol diet[42].

In summary, our study highlights the importance of including diet in experimental setups for understanding molecular mechanisms associated with complex traits and suggests that the same should be done in human GWAS to avert spurious associations. In terms of clinical translation, identifying gene-environment interaction may help to identify pharmaceutical interventions that are beneficial for a defined subgroup of the population carrying a specific genotype[43]. For instance, it is tempting to speculate, based on the results of our study that lupus patients expressing lower levels of the TNXB gene are likely to

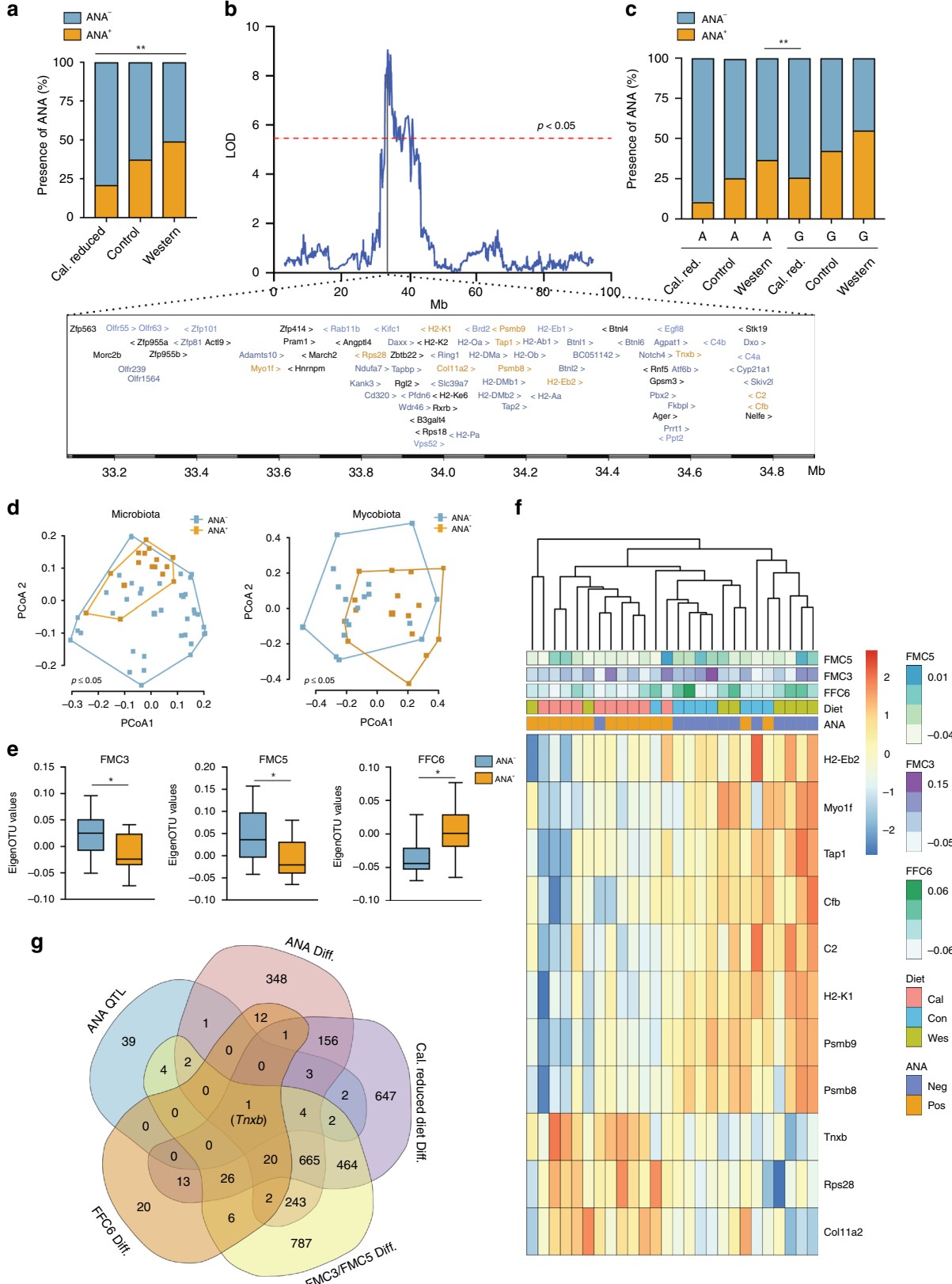

benefit more from caloric restriction. Moreover, our results in the NZM2410/J mice indicate that dietary regulation of the microbiome is associated with lupus development, suggesting that dietary interventions and/or use of probiotics may be used as preventive measures in populations at risk.

## Methods

**Animal experiments**. The four-way advanced intercross line was generated by intercrossing MRL/MpJ, NZM2410/J, BxD2/TyJ and Cast/EiJ strains at equal strain and sex distribution. Mice were intercrossed for 20 generations with at least 50 breeding pairs per generation. Offspring mice used in this study were transferred into separate cages after weaning at 3–4 weeks. Each cage contained mice of either

**Fig. 5** Fine mapping of diet-associated ANA QTL. **a** Stacked percent bar plot shows distribution of ANA between three dietary groups caloric restriction ($n_{ANA-} = 239$, $n_{ANA+} = 64$), control ($n_{ANA-} = 319$, $n_{ANA+} = 192$) and western diet ($n_{ANA-} = 149$, $n_{ANA+} = 145$) in AIL mice. **b** The X-Y line plot depicts the log of the odds (LOD) ratios on the y-axis and positions on chromosome 17 (Mb) on the x-axis with the genome-wide threshold indicated in red. The insert below shows genes present in the ANA QTL. While genes highlighted in blue contain consequential variation derived from the genome of the NZM2410/J mice, genes shown in orange are also differentially expressed between ANA-positive and ANA-negative mice. **c** Stacked percent bar plot illustrating the distribution of $ANA^+$ ($n = 401$) vs. $ANA^-$ ($n = 707$) mice in the AIL population based on the allele within the peak SNP (UNC27797171; $n_{AA} = 104$, $n_{AG} = 459$ and $n_{GG} = 545$) and diet ($n_{calorie\ restricted} = 303$, $n_{control} = 511$ and $n_{western} = 294$). While the A allele is derived from NZM2410/J mice, the G allele is derived from the MRL/MpJ, BxD2/TyJ, or CAST/EiJ mice. Data for panels a and c were analyzed using the $X^2$ test with Fischer exact test as a post hoc test. $*p < 0.05$, $**p < 0.01$, n.s., not significant. **d** PCoA plot showing differences in beta diversity in the microbiome (unweighted UniFrac distance, left panel) and mycobiome (Jaccard distance, right panel) between ANA positive ($n_{microbiome} = 16$, $n_{mycobiome} = 16$) and negative samples ($n_{microbiome} = 37$, $n_{mycobiome} = 18$). Statistical significance was assessed using the adonis function in R (permutations = 999). **e** Boxplot (the band indicates the median, the box indicates the first and third quartiles and whiskers indicate 1.5*interquartile range) illustrates significant functional microbial (FMC3 & FMC5) and fungal (FFC6) communities associated with ANA presence ($n_{FMC} = 16$, $n_{FFC} = 16$) or absence ($n_{FMC} = 37$, $n_{FFC} = 18$) (Mann–Whitney $U$ test, $*p < 0.05$). **f** Heatmap showing the expression of genes that are differentially expressed for the ANA phenotype and present within the ANA QTL. **g** Five-way Venn diagram demonstrating a multi-omics strategy used to fine map a diet-regulated gene within the ANA QTL. Source data for (**a**, **c**–**f**) are provided in the Source Data file

---

sex and was randomly allocated to one of the three different diets: control mouse chow, caloric restriction, and Western diet. Control mouse chow (#1320, Altromin Spezialfutter GmbH, Lage, Germany) was given ad libitum. Caloric restriction was performed by 40% reduction of the control mouse chow consumed by the age and sex matched controls. Western diet is rich in cholesterol, butter fat, sugar (S0587-E020, ssniff Spezialdiäten GmbH, Soest, Germany). All 1154 animals were held under specific pathogen-free conditions at 12-h light/dark cycle at the animal facility of the University of Lübeck, Germany. The mice were kept on the corresponding diets until the age of 6 months, at which point 1154 animals were still alive. At the age of 2- and 4-months blood collection by facial vein puncture was performed. Furthermore, stool samples were collected and the current weight of the animals was obtained on the 2nd, 4th, and 6th months of dietary intervention. At the age of 6 months all animals were euthanized and sampled for stools, blood, and liver. All animal experiments were conducted according to the European Community rules for animal care, approved by the respective governmental administration (Ministry for Energy, Agriculture, the Environment and Rural Areas, file number 27–2/13) and performed by certified personnel.

**Differential blood cell counts**. Blood was collected at 2 and 4 months of age by facial vein puncture. At 6 months of age, mice were euthanized using $CO_2$ and blood was collected by a cardiac puncture. Immediately after collection, 20 µl of whole blood were added into a tube containing 20 µl of EDTA. The measurement of whole blood samples was performed using HemaVet 950 (Drew Scientific Inc, Miami Lakes, USA). After the measurement, the corresponding results were adjusted to the dilution factor.

**Liver staining and analysis**. Histological analysis of the obtained liver samples was performed on 10 µm hematoxylin/eosin stained formalin-fixed sections. The scoring was performed by an investigator unaware of the dietary allocation of the mouse according to standard protocols. Lobular inflammation and hepatocyte ballooning were evaluated semi-quantitatively: steatosis (0–3), lobular inflammation (0–2), hepatocellular ballooning (0–2), and fibrosis (0–4); whereby 0 indicates the absence of the items and higher numbers correlate to an increasing severity.

**HDL/LDL analysis**. Prior to blood collection, mice were fasted overnight. After blood collection, serum samples were obtained by centrifugation at $13,000 \times g$ for 10 min. Total cholesterol and HDL/LDL were measured using the EnzyChrom AF HDL and LDL/VLDL Assay Kit (BioAssay Systems, Hayward, USA) according to the manufacturer's instructions.

**Glycosylation**. Total murine IgG glycosylation was measured by HILIC-HPLC. Total IgG from murine serum was purified by Protein G coupled monolith material (BIA Separations, Vienna, Austria). The Fc N-glycan was then released by endoglycosidase S (EndoS) derived from Streptococcus pyogenes that cleaves the glycan in the chitobiose core. The glycans were then labeled with anthranilamide (Sigma-Aldrich GmbH, Darmstadt, Germany) and separated by hydrophobicity on a Dionex Ultimate 3000 HPLC (Thermo Fisher Scientific GmbH, Dreieich, Germany) using a Xbridge XP BEH Glycan column (1.7 µm, 100 × 2.1 mm i.d.; Waters, Milford, MA USA). Peaks for agalactosylated (G0), mono- (G1) or bi-(G2) galactosylated as well as mono- (G1S1 or G2S1) and bi-(G2S2) sialylated glycans were detected. Bisected glycans were not observed. The percentage distribution of the detected peaks was calculated by dividing the area under the curve of a specific peak with the total area of all detected peaks. In addition, the following traits were derived from the detected glycans: "Gal" (all peaks except G0), "term gal" (peaks G1 + G2), "sial" (peaks G1S1 + G2S1 + G2S2), "sial by gal" ("sial" divided by "gal"), "mono gal" (peaks G1 + G1S1), "bi gal" (peaks G2 + G2S1 + G2S2).

**Immunoglobulins**. Total immunoglobulin IgG, IgA, IgM isotypes levels were measured in serum samples by sandwich ELISA (Bethyl Laboratories, Montgomery, TX, USA), according to manufacturer's instructions. In total, serum samples from 534 mice were measured. Also, six additional traits were calculated as linear combinations of the log-transformed and standardized individual immunoglobulin isotype values (IgA, IgG and IgM)[44]. For general immunoglobulin production capacity, we calculated total immunoglobulin AGM as $(\log(IgA) + \log(IgG) + \log(IgM))$. The efficacy of class switching was defined by AG $(\log(IgA) + \log(IgG))$, while the ratio of class-switch to non-class-switch immunoglobulins was measured as AG/M $((\log(IgA) + \log(IgG)) - \log(IgM))$. Isotype-specific class switching was calculated by A/M $(\log(IgA) - \log(IgM))$ and G/M $(\log(IgG) - \log(IgM))$. The opposite direction on the two isotypes was captured by A/G $(\log(IgA) - \log(IgG))$.

**CRP measurement**. Mouse CRP was measured in mouse serum using mouse CRP DuoSet ELISA (R&D Systems, Wiesbaden-Nordenstadt, Germany) in accordance with the manufacturer's instructions. The CRP concentration was determined in 435 serum samples at 6 months after they were set on different diets.

**Indirect immunofluorescence microscopy on HEp-20–10 cells**. For the detection of circulating ANA, serum samples of AIL and NZM2410/J mice were investigated using indirect immunofluorescence on HEp-20–10 cells (Euroimmun AG, Lübeck, Germany). Briefly, serum samples were diluted 1:100, 1:1,000 and 1:10,000 in PBST, added to HEp-20–10 Biochips (Euroimmun AG, Lübeck, Germany) and incubated for 30 min at room temperature. Afterwards, slides were washed twice for 5 min with PBST and treated with 1:100 diluted FITC-conjugated polyclonal swine anti-rabbit IgG (Dako, Hamburg, Germany). The analysis of samples was performed by an investigator who was unaware of the dietary allocation of the mice.

**Genotyping of AIL mice**. Genomic DNA was isolated from the tips of the tails of AIL mice obtained during mouse sampling at month 6. Purification was performed using the DNeasy Blood & Tissue Kit (Qiagen GmbH, Hilden, Germany) according to the manufacturer's protocol. Extracted DNA was quantified using NanoDrop (Implen, Munich, Germany) and adjusted to 50 ng/µl in TE Buffer (10 mM Tris, 1 mM EDTA, pH 8). DNA samples were stored at −20 °C until further use. DNA samples from 1154 mice were analyzed by MegaMUGA genotyping array covering 77,800 markers throughout the mouse genome. Genotyping was performed at Neogen/GeneSeek (Lincoln, NE). Using plink, we filtered out noninformative SNPs based on minor allele frequency (maf) > 0.05, missing geno probability < 0.1 and common homozygous SNPs among the founders resulting into 55,458 SNPs, which were used in downstream analysis[45]. We used HAPPY R package for probabilistic reconstruction of AIL mouse genome in term of four founder strains[46]. Using a hidden Markov model (HMM), at every adjacent marker interval across a chromosome, we estimated the posterior probability that each mouse was in one of the four possible genotype states. These probabilities were converted to three dimensional arrays in R. and A kinship matrix, which represents intra-individuals relationship, was calculated using kinship.probs function (DOQTL R package)[47].

We applied box-cox transformation to all the quantitative traits to bring it to normal shape except body weight which was already normally distributed. Afterwards, we fitted each trait for sex, diet as fixed effect and kinship as random effect to estimate residuals (r) using hglm R package[48]. An advantage of such approach is that, (i) in hglm models distributions other than gaussian (such as binomial), can be incorporated and (ii) it also results in increased performance of the downstream statistics as computationally expensive high dimensional kinship matrix regression is not required while associating residuals from the traits to the genotype while conducting permutation to access significance.

In our study, we tested three types of model for identification of host genetics loci (QTL) associated with studied traits. (i) host-genotype (G) expressed as posterior probability from four founders association with r (residuals from hglm). In this model, we calculated log likelihood ratios of traits for each interval across the genome and converted them to log of odd ratios (LOD scores). (ii) G × Diet association with r. In this model, LOD scores were calculated by comparing trait log likelihood for G with G × Diet[49] (iii) Similarly, for G × Sex interactions LOD scores were calculated by comparing trait log likelihood for G with G × Sex. Genome (significant) and chromosome (suggestive) wide significance, denoted as $\alpha_{g\text{-}w}$ and $\alpha_{c\text{-}w}$, as estimated by traditional permutation (1000) for each trait-based method at 5% threshold. Off note, if the QTL within the chromosome was significant for genome-wide threshold than the QTL was not considered as chromosome-wide (suggestive) QTL. The confidence interval for a QTL was described by 1.5 LOD drop.

**Whole-genome sequencing of founder strains**. Genomic DNA of three founder strains (NZM2410/J, MRL/MpJ, BxD2/TyJ) was isolated from the tips of the tails of mice from each of the founder strains using the DNeasy Blood & Tissue Kit (Qiagen GmbH, Hilden, Germany) according to the manufacturer's instructions. To detect the possible DNA degradation, the quality of the obtained genomic DNA was controlled by electrophoresis on a 0.7% agarose gel at 15 V overnight. Whole-genome sequencing was performed on a HiSeq X machine, 150 × 2 paired end sequencing with ~30× (Quick Biology, Pasadena, CA, USA). Data were obtained in FastQ format. The quality of the sequenced reads was evaluated by Fastqc software[50], and reads with phred score < 30 were filtered. The remaining reads were aligned to C57BL/6 J GRCm38 (mm10) mouse reference genome and BAM file per strain was obtained using the BWA-MEM (v0.7.10) software with default parameters[51]. The BAM files were evaluated for quality of the alignment with reference genome using Qualimap software[52]. Downstream analysis for SNP and indel detection was performed as following. Each BAM file was sorted and filtered for possible PCR and optical duplicates using Picard Tools (v1.141). Reads were realigned to improve SNP and indel calling around indels using default options by the GATK v3.5 'IndelRealigner' tool. We used a combination of SAMtools mpileup (v1.5) and BCFtools call (v1.5) for the identification of SNPs and indels. The following options were specified for SAMtools: '-t DP,DV,DP4,SP,DPR,INFO/DPR -I -E -Q 0 -pm 3 -F 0.25 -d 500 -ug' and BCFtools call; '-mv -f GQ,GP -p 0.99'. To improve the accuracy of indel calls, indels were then left-aligned and normalized using the bcftools norm function with the parameters '-D -s -m + indels'. To ensure high quality, we used bcftools annotated with the following parameters: "StrandBias = 0.0001, EndDistBias = 0.0001, MaxDP = 150, BaseQualBias = 0, MinMQ = 20, MinAB = 5, Qual = 10, VDB = 0, GapWin = 3, MapQualBias = 0, SnpGap = 2, MinDP = 5" was used as a soft filter for SNPs and indels variants. These filtered out low confidence variants and removed false positive SNPs and indels due to alignment artifacts. We retained only high-quality (i.e., passed all filters) and homozygous variants. Additionally, the SNPs and indels common among the four founders were filtered out. SNPs and indels were annotated for their functional class, consequence and known transcripts if available using Ensembl VEP[53].

**Strategy for fine-mapping QTL using whole-genome sequencing**. We mapped distinct complex traits to various chromosomal regions (QTL) of mouse genome within confidence interval of 2–3 Mb (Supplementary Data 1). We further aimed to resolve these regions to single or few genes using whole-genome sequencing of founder strains. Briefly, for each QTL we estimated the founder allele effect. In most cases, the higher LOD score is a consequence of differences of one or two strains alleles from other founders, due to single diallelic polymorphism leading to variation in traits. These differences were manually inspected and strain-specific SNPs and Indels were kept in a given QTL. Further, we prioritized the identified SNPs and Indels based on their consequences (Ensembl VEP). For example, we filtered all the synonymous SNP for the downstream analysis. This approach led to filtering of the genes which were not polymorphic among the founders and therefore inconsequential for the variation in the trait. The remaining genes in the QTL were investigated for association with traits by curated databases such as GWAS Catalog[54] and GeneCARD[55] and thorough literature search by two independent investigators. Once we identified several traits associated genes we investigated SNPs and Indels associated to the genes. We prioritized polymorphisms in 5′UTR, 3′UTR (regulator of gene expression) and missense mutation over other type of consequences.

**Dietary intervention in NZM2410/J**. All animal experiments were approved by the Ministry for Energy, Agriculture, the Environment and Rural Areas, file number 35–3/10 and performed by certified personnel. The NZM2410/J breeding pairs were obtained from the Jackson laboratories and further breeding was performed in the animal facility of University of Lübeck, Gemany. A total of 55 NZM2410/J mice were randomly allocated to one of the three diets, (control, Western or caloric restriction) after weaning at 3 weeks of age. Control diet (S0587-E001, ssniff Spezialdiäten GmbH, Lage, Germany) was given ad libitum. Caloric restriction was performed by 40% reduction of food amount consumed by sex and age matched control diet mice. Western diet was rich in cholesterol, butter fat,

sugar (S0587-E020, ssniff Spezialdiäten GmbH, Lage, Germany). A detailed overview of the exact composition of each of the diets can be reviewed in Supplementary Table 3. The mice were kept on the respective diet for 28 weeks. Blood was collected monthly by facial vein puncture, stool samples from individual mice were collected every 2 weeks and proteinuria was measured weekly. Mice that developed clinical disease were euthanized in the event that they lost 25% of their body weight. All surviving animals were euthanized and sampled for stool, blood, kidneys, and spleens.

**Proteinuria analysis**. Urine samples were collected weekly by gentle urinary bladder massage. The samples were tested using Combur3 Test urine sticks (Roche, Mannheim, Germany). If proteinuria above 100 mg/dl was detected on two consecutive weeks, the date of the first detection was determined as clinical manifestation of proteinuria.

**Gene expression profiling of NZM2410/J spleen**. Total RNA was extracted from frozen spleen samples preserved in RNAlater using TRIzol™ reagent (Thermo Fischer Scientific GmbH, Dreieich, Germany) according to the manufacturer's instructions. Total RNA was further treated with a DNase I kit (Qiagen GmbH, Hilden, Germany) and purified with columns using a QIAamp RNA Blood Mini Kit (Qiagen GmbH, Hilden, Germany) in accordance with the manufacturer's instructions. Consequently, RNA concentrations were quantified using a Nano-Drop 2000c spectrophotometer (Thermo Fischer Scientific GmbH, Dreieich, Germany). cDNA libraries were constructed using Illumina's TruSeq® RNA Sample Preparation V2 kit (Illumina Inc., San Diego, CA USA) following the procedures outlined in the manufacturer's manual and sequenced on Illumina NextSeq machine in a High Output mode. Bcl files were obtained from the Illumina Nextseq system and converted to fastQ using CASAVA's bcl2fastq2 (v.2.19). The fastQ paired end reads were trimmed using Trimmomatic (v.0.36) for removing adapter sequences and low quality ($q < 20$) sequences[56]. We used Tuxedo protocol for downstream processing of RNAseq data[57]: The filtered fastQ reads were aligned to the mouse genome (GRCm38) using tophat2 (v2.0.13) with the default parameter, and bam files were generated for each sample. Cufflinks (v2.2.1) were used to assemble transcripts, and a gtf file per sample was created. Cuffmerge with reference known gene GTF (downloaded from Ensembl database) was used to merge assemblies into a single GTF file for all the samples. The abundance or counts of every known and novel transcript across samples was summarized using feature-Counts (v1.5.2)[58]. The output was used as input to the DESeq2 R package[59]. In the DESeq2 R package, differentially expressed genes were identified by the Wald test for binary traits, such as disease and ANA, and the likelihood-ratio test for multiple groups, such as diet. The identification of differentially expressed genes corresponding to microbiota and mycobiota was performed using eigenOTU values derived for the modules turquoise (microbiota) and FFC4 (mycobiota) as quantitative traits. The genes with $Padj < 0.05$ were considered significant. The enriched reactome pathways and gene networks were identified using gene ontology consortium and INMEX web server[60,61].

**16S rRNA gene sequencing and analysis**. The hypervariable V1–V2 region of the bacterial 16S rRNA gene was amplified following a dual indexing approach for each sample following standard protocolls[62]. All primers used in this study are indicated in Supplementary Table 4. In brief, the primers used for amplification contain universal bacterial primers 27F and 338R as well as P5 (forward) and P7 (reverse) sequences

(5′AATGATACGGCGACCACCGAGATCTACACXXXXXXXXXTATGGTAAT TG

TAGAGTTTGATCCTGGCTCAG-3′) and (5′-CAAGCAGAAGACGGCATA CGA

GATXXXXXXXXXAGTCAGTCAGCCTGCTGCCTCCCGTAGGAGT-3′).

To increase annealing temperature of the sequencing primers, as recommended, a 12-base linker sequence was added to bacterial primer. Each PCR product was tagged using unique eight-base multiplex identifier, included in the primers (designated as XXXXXXXX). Total RNA was extracted using the AllPrep DNA/ RNA Qiagen kit (Qiagen GmbH, Hilden, Germany) according to the manufacturer's protocol. cDNA synthesis was performed using High-Capacity cDNA Reverse Transcription Kits (Thermo Fischer Scientific GmbH, Dreieich, Germany). The purity of the isolated RNA was evaluated by negative reverse transcriptase PCR and agarose gel electrophoresis. PCR amplifications were done using the cDNA template (100 ng in a volume of a 12.5 μL) using the Phusion® Hot Start II DNA High-Fidelity DNA Polymerase (Finnzymes, Espoo, Finland). Cycling conditions were as follows: initial denaturation for 30 s at 98 ℃; 30 cycles of 9 s at 98 ℃, 30 s at 55 ℃, and 30 s at 72 ℃; final extension for 10 min at 72 ℃. Reactions were duplicated and products were merged in order to obtain a final volume of 25-μL PCR for each sample. To check the purity of the PCR products, blank (template-free) reactions using different combinations of forward and reverse primers were added. Next, using an image analysis software (Bio-Rad Laboratories GmbH, Munich, Germany), we quantified PCR product concentrations, and products were further pooled to generate equimolar subpools. Subpools were then extracted from agarose gel with the Qiagen MinElute Gel Extraction Kit (Qiagen GmbH, Hilden, Germany) and quantified with the Quant-iT™ dsDNA BR Assay Kit on a Qubit

fluorometer (Thermo Fischer Scientific GmbH, Dreieich, Germany). Ultimately, for each library, subpools were combined to one single equimolar pool. Further purification was achieved using AMPure® Beads (Beckman Coulter, Brea, CA, USA). As recommended by the supplier, prior to sequencing, libraries were run on an Agilent Bioanalyzer. Finally, using the MiSeq Reagent Kit v3 600 cycles sequencing chemistry (Illumina Inc., San Diego, CA, USA), these libraries were sequenced on a MiSeq.

No mismatch to the barcode was allowed while demultiplexing (CASAVA, Illumina). USEARCH was used to merge raw forward and reverse reads USEARCH (v.7)[63]. In USEARCH, both forward and reverse reads were trimmed where first base below quality score of $Q = 2$ was found. The trimmed reads were used for merging paired ends, where we allowed minimum two mismatches in the overlapping region of minimum 150 bp and minimum read length of 200 bp. Merged reads with length of less than 200 bp or more than 330 bp were discarded from downstream analysis. These reads were also filtered by parameter of expected error ($E = 0.5$). Chimeric sequences were removed by both the de novo method (UCHIME)[64] and the reference-based method (comparison with the SILVA Gold reference database). We used the RDP classifier (v11.0) at confidence 0.80 and 1000 iterations to classify sequences as Phylum to Genus level[65]. Species level OTUs (operational taxonomic units) classification was performed representative unique FASTA sequences for each OTU derived was using the usearch algorithm at a threshold of 0.97 (binning at 97 % similarity)[63]. Taxonomy was assigned to each OTU by comparing with the Greengenes database using uclust[66]. For downstream analysis, we constructed a phylogenetic tree using various functionalities of QIIME software[67]. First, we used representative fasta sequences for each OTU and aligned it against the Greengenes database using script align_seqs.py with the pyNAST algorithm. The filter_alignment.py was used to remove positions that are gaps in every sequence. Finally, a phylogenetic tree between the represented OTU sequences was constructed using the fasttree algorithm (make_phylogeny.py).

**Quantitative and ecological analysis**. As reported earlier, for ecological analysis, we removed singletons and subsampled our OTU data to 20000 reads per sample. An alpha diversity (α) index (Chao1 index) was calculated for each stage separately with a QIIME package[67]. The difference in α diversity for dichotomous traits such as disease state (continuously healthy or future/present disease) or sex (female/male) was assessed by the Mann–Whitney $U$ test and polychotomous variables such as diet at each stage separately (i.e., naive, transient and final) by the Kruskal–Wallis test. To measure beta diversity, we calculated the unweighted UniFrac distance matrix among the samples. Using a distance matrix, PCoA (principle coordinate analysis) was performed to compare different groups. The significance among the groups was assessed by the adonis R function with 999 permutations. To identify differentially abundant taxa (phylum to genus by RDP classifier and species level OTUs) across the groups, we used the LEfSe (Linear discriminant analysis effect size) algorithm[29]. In this analysis, taxa were considered significant when the LDA (linear discriminant analysis) score was >1.5 and the Mann–Whitney $U$ test (disease) or Kruskal–Wallis test (diet) $P$-value was <0.05. We estimated the FMC using python-based SparCC (Sparse Correlations for Compositional) and R-based WGCNA program[68]. We first filtered OTUs with a minimum of 10 reads in at least 25 samples, resulting in 293 OTUs for downstream processing. Afterwards, we used SparCC ($\rho_{ij}$) to infer the co-occurrence relationship between the OTUs. Then, a weighted adjacency matrix (network) was defined by raising $\rho_{ij}$ to a power $a_{ij} = (0.5 + 0.5\rho_{ij})^\beta$, with $\beta = 4$[32]. We identified clusters of FMCs based on the topological overlap measures of adjacency matrix derived from branches of hierarchical clustering tree. The minimum cluster size used was 10. To summarize the profiles of co-occurrence modules, eigenOTUs were computed as implemented in the R function module Eigengenes. These eigenOTUs were correlated with different traits, such as the presence or absence of proteinuria, diet, stages and sex, to obtain a module-trait relationship. Additionally, the correlation among the species level OTUs between functional microbial and mycobial community, fungal-fungal interaction, bacteria-bacteria interaction and fungal-bacterial interaction was calculated by SparCC implementation in R (FastSpar), and p-values were adjusted for multiple comparison by the Benjamini-Hochberg procedure[68].

**ITS2 gene sequencing and analysis**. An internal transcribed spacer region 2 (ITS2) gene library was constructed according to a dual indexing strategy using the fITS7 (forward) and ITS4 (reverse) primers (forward 5′- AATGATACGGCGAC CACCGAGATCTACACXXXXXXXXXTATGGTAATTG

GTCCTCCGCTTATTGATATGC-3′, reverse 5′-CAAGCAGAAGACGGCATAC GAGATXXXXXXXXXAGTCAGTCAGCCGTGA[AG]TCATCGAATCTTTG-3′)[69] (Supplementary Table 4). The primers contain a unique multiplex identifier (designated as XXXXXXXX), the 10-nt pad sequence to prevent hairpin formation (underlined), the 2-nt linker sequence, and ITS2-specific primer sequences. The reverse primer was degenerated at one position. The stool samples were stored in RNAlater (QIAGEN, Hilden, Germany) in a −20 °C freezer before DNA isolation. RNAlater was removed by washing the samples twice in PBS. Afterwards, DNA was extracted using the DNeasy PowerLyzer PowerSoil Kit (Qiagen GmbH, Hilden, Germany) according to the manufacturer´s instructions with an additional step of 2 h incubation at 55 °C with Proteinase K (Qiagen GmbH, Hilden, Germany), followed by homogenization at 6000 rpm 3 × 15 s in Precellys tissue homogenizer (Bertin instruments, Frankfurt am Main, Germany). PCR amplifications were

conducted using the Phusion® Hot Start II DNA High-Fidelity DNA Polymerase (Finnzymes, Espoo, Finland). Cycling conditions were as follows: initial denaturation for 30 s at 98 °C; 35 cycles of 9 s at 98 °C, 30 s at 50 °C, and 30 s at 72 °C; final extension for 10 min at 72 °C. Template-free reactions using different combinations of forward and reverse primers served as negative controls. PCR product concentrations were quantified on a 1.5% agarose gel using CAPT-analysis software (Vilber Lourmat, Marne-la-Vallée, France). Following quantification, products were mixed together to make equimolar subpools. Subpools were then extracted from agarose gel with the Qiagen MinElute Gel Extraction Kit (Qiagen GmbH, Hilden, Germany) and quantified with a Qubit dsDNA HS assay kit on a Qubit fluorometer (Thermo Fischer Scientific GmbH, Dreieich, Germany). Subpools were then combined in one equimolar pool to a single library. The final library was purified using Agencourt AMPure® Beads (Beckman Coulter, Indianapolis, USA), quantified by a NEBNext Library quantification Kit (New England BioLabs GmbH, Frankfurt am Main, Germany) and subjected to analysis on the Agilent Bioanalyzer (Agilent, Santa Clara, CA, USA) prior to sequencing. The Amplicon libraries were sequenced on a MiSeq using the MiSeq Reagent Kit v3 600 cycles sequencing chemistry (Illumina Inc., San Diego, CA, USA). Similar to microbiota, no mismatch to the barcode was allowed while demultiplexing mycobiota reads (CASAVA, Illumina).

**Quantitative and ecological analysis**. The raw mycobiota data were processed using PIPITS pipeline[69]. Briefly, the paired end reads were merged and filtered using the "PIPITS_PREP" module. The fastx-toolkit was used to merge paired reads, and reads below $q < 20$ were filtered out. Next, "PIPITS_FUNITS" was used to extract reads that belong to ITS region. We used the UNITE UCHIME dataset for reference-based chimera removal and UCHIME for de novo chimera removal by the vsearch algorithm (v.2.8) with $E = 0.5$[70]. Phylum to genus level classification of the reads was performed using RDP classifier with UNITE database as reference at confidence 0.80 and 1000 iterations[71]. OTU clustering at 97% threshold was performed using vsearch with default parameters. The taxonomic classification of the FASTA sequences for each OTU was performed using the RDP classifier. The abundance table was extracted for both taxa classification and OTUs after removing singletons and used in QIIME for ecological analysis. The samples were divided into three stages and subsampled to 10,000 reads. The samples that did not pass the threshold were eliminated from the study. The α-diversity was obtained using the Shannon index (species evenness), and significance was assessed using the Mann–Whitney $U$ test or Kruskal–Wallis test. For beta diversity, Jaccard distance (nonphylogenetic distance) was calculated among the samples, and PCoA was performed. The significance was assessed by the adonis function in R. With the same parameter as in microbiota, we identified differentially abundant mycobiota taxa using LefSe[29]. Similar to FMC, we also calculated FFC (functional fungal community). For clustering modules, the parameters were kept the same except for the minimum cluster size, which was kept at 5 due to the lower diversity of fungal species. The eigenOTUs from FFCs were correlated with different traits, such as sex, diet, disease and stage, to identify FFCs associated with each trait.

**Reporting summary**. Further information on research design is available in the Nature Research Reporting Summary linked to this article.

## Data availability

Whole-genome sequencing data for BxD2/TyJ, MRL/MpJ, and NZM2410/J mice strains have been deposited in database European Nucleotide Archive (ENA) in FASTQ format and publicly available under accession number [PRJEB29771]. The raw sequencing data, i.e., FASTQ files for RNA-Seq, microbiome and mycobiome from NZM2410/J, have been deposited in public database NCBI SRA under accession number [PRJNA543200]. Additionally, Plink formatted genotype data (bed and bim files) for advance inter-cross line mice, quality control of alignment from whole-genome sequencing (Qualimap output), VCF files from sequenced strains and founder coefficient plots for every genome-wide QTL are publicly available on the Dryad database [https://doi.org/10.5061/dryad.c8gc64n]. The data can be visualized and explored at [http://diet.ag-ludwig.com]. The source data underlying Figs. 1a, 2b–c, 2e–g, 3a–f, 4a–d, 5a, 5c–f and Supplementary Figs. 1a–c and 2a–h, 3a–c, 4a–b are provided as a Source Data file. All other data supporting the findings of this study are contained within the article and its Supplementary information files.

## Code availability

All codes generated or used during the current study are available at Github repository and Zenodo database [https://doi.org/10.5281/zenodo.3347025].

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

## Acknowledgements

The study was supported by the Excellence Clusters "Inflammation at Interfaces" (EXC 306) and "Precision Medicine in Chronic Inflammation" (EXC 2167–390884018), as well as the Research Training Groups "Genes, Environment and Inflammation" (GRK 1743) and "Modulation of Autoimmunity" (GRK 1727), all from the Deutsche Forschungsgemeinschaft. We would like to thank Claudia Kauderer for an excellent technical assistance, Andreia Marques for maintenance of murine lines, and Silke Carstensen who extracted nucleic acids and generated the 16S rRNA libraries. We further would like to acknowledge the computational support from the OMICS Compute Cluster at the University of Lübeck (https://www.itsc.uni-luebeck.de/dienstleistungen/omics.html).

## Author contributions

A.V., Y.G., T.S., and R.J.L. designed the study and the experiments, and wrote the manuscript. A.V. and H.K. conducted the animal experiments. Y.G., with support from S.M., conducted the bioinformatical analysis throughout the study. T.S., A.V., P.S., A.L. E., F.B., and C.D.S. generated the mycobiota data. T.S., C.D.S., and S.K. generated the RNA-Seq data. H.K.A. determined HDL and LDL levels. P.K. evaluated ANA. Y.C.B., M.S., and M.E. acquired and analyzed the glycosylation data. S.K., M.B., and J.F.B. generated the microbiome data. F.B., J.J., S.Kh., and K.B. quantified CRP and immunoglobulins. H.A. and C.S. phenotyped the mice for NAFLD. A.V., H.N., and T.N.M. performed the histopathology of kidneys of the NZM mice. R.A.M., S.D., W.H.B., D.Z., and S.M.I. contributed to the design of the study and writing of the manuscript.

## Additional information

**Competing interests:** R.J.L. has received research funding from Miltenyi Biotec, Biogen, Biotest, Almirall, True North Therapeutics, UCB Pharma, ArgenX, TxCell, Topadur, Incyte and Admirx and fees for consulting or speaking from ArgenX, Immunogenetics, Novartis and Lilly. All other authors declare no competing interests.

