## [Peer Review File · Nature Communications]

Reviewers' comments:

Reviewer #1 (Remarks to the Author):

Vorobyev et al. describe a large and in-depth characterization of the impact of diet on metabolic and immune function in a genetically diverse mouse population. They demonstrate that diet induces profound changes in the trait-specific genetic landscape. They further demonstrate changes in micro- and myco-biota due to diet. While diet induced changes in microbiome are not surprising, the authors demonstrate that these changes occur prior to and predict the onset of autoimmune disease. The wide range of traits impacted by diet is impressive. The mouse population is large and the study appears to be well powered for high resolution mapping. Indeed, many QTL are associated to specific genes that include well established, plausible, and novel candidates.

Major concern:

I may have overlooked but it does not appear that the study data have been deposited in a public repository, aside from the deposition of the genomic sequencing data. A study of this type cannot be replicated/validated without access to the source data and, ideally, a documentation of the computer code used in the analysis. Public deposition of the data would greatly increase the value of the study and likelihood of data re-use and further discovery. Recommend Dryad or Mouse Phenome Database.

Additional minor concerns:

The opening paragraph of the introduction makes broad claims about heritability that seem to confound concepts of environmental and genetic effects. A more thoughtful discussion of how GxE could impact heritability estimation, with some up to date citations (e.g. work by Noah Zaitlen), would be helpful. But the work presented does not address the missing heritability problem. For example, there are no estimates of heritability or genetic variance reported for their study population. The few references provided are all from 2009 or earlier. Likewise the discussion of genetic of mouse populations is outdated and narrow in scope (two old references from the same group of researchers). In this vein, the title makes too bold of a claim and does not accurately reflect the content of the paper.

The authors use the term "gender" when they actually mean "sex". Gender is a social/cultural construct whereas sex refers to the biological state of an organism. Mice do have gender but the authors clearly mean sex.

Reviewer #2 (Remarks to the Author):

Major comments:

Vorobyev et al. submitted a manuscript thoroughly investigating missing heritability of complex metabolic and immunological traits using an autoimmunity-prone advanced mouse intercross. The authors address a very relevant and widely discussed question, why only a small proportion of phenotypic variation is explained by host genetics when QTL mapping is performed. The authors hypothesized that diet as a main environmental factor could be a key regulator affecting complex traits and the explanation of missing heritability.

In this study, the authors fed more than 1100 mice different diets and showed that diet contributes

not only to the variability in complex phenotypic traits in metabolism and immunological functions but also contributed to the discovery of new QTLs associated with genetic susceptibility that were so far overlooked.

Altogether, the authors present a study that impressed me due to a stringent and comprehensive design including the analysis of the microbiom and even mycobiom as possible disease mediators, the finding that diet may overcome genetic susceptibility, and the identification of Tnxb as a new candidate gene based on multi-omics approaches. Even though the study still describes mainly associations but not real causality these findings are relevant and suggest functionality.

Further comments:

1. Diet treatment: The treatment is generally described in the supplement material. Two aspects should be clarified: a) 40% calorie restriction based on the ad libitum uptake of sex and aged-matched mice with higher body weight presumably does not affect energy balance too much especially at a later age stage. Mice likely compensate by reducing locomotor activity and metabolic rate at resting. Could that have an impact on the results? b) I could not find detailed information on the Western diet. Regarding reproducibility I would recommend a detailed description of the experimental diets including details if the diet was a grain-based chow or a purified diet which would have considerable effects on the microbiome and likely also on metabolic functions. Regarding this context, I wonder if the authors were mainly interested in calorie consumption or specific diet composition (e.g. higher fat content). What was the experimental rationale?
2. Personally, I still prefer the term "sex" as a main experimental factor in an animal study instead of "gender".
3. L 344 - LEfSe algorithm – I am not sure if every reader is familiar with that term – brief explanation?
4. L 358 - Why "unceasingly" – is this term justified?
5. L 359 – "presence or absence of certain microbial species precedes the onset of the disease phenotype" – maybe a bit simplistic for complex pathophysiological changes over time.
6. L 382 "FMCs" – very brief explanation may be helpful
7. L 382 "eigenOTUs" see above – very brief explanation may be helpful
8. In the discussion, I missed that the factor "sex" is not discussed in more detail.
9. Overall, the data provide evidence for functional links but the study is still about associations and not causality. I think that needs to be stressed at some point in the discussion.
10. Can the authors briefly discuss the translational value of their study?

Reviewer #3 (Remarks to the Author):

The authors submit a manuscript entitled, "Diet bridges the missing heritability." This manuscript utilizes an advanced intercross line fed different diets and show that diet interactions change the genetic associations with a plethora of physiologic traits measured. The next part of the paper is focused on the NSM2410/J mice and detailed phenotyping the mice on different diets. This is followed

by microbiome analysis and finally the paper finishes with a candidate gene associated with a candidate gene associated with ANA phenotypes.

1. One major flaw in the general mouse genetic study with the AIL is that each mouse is unique and while you observe differences in phenotypes you have no way to know truly how the diet effects genetics because diet can not be compared across unique mice. While one could argue that the alleles are constantly recycling in the AIL, it would still make analysis of diet interactions quite difficult. How do the authors reconcile this fact? There are quite a few mouse genetics studies that show similar effects.

2. I am concerned about that the high-resolution mapping is greatly overstated. The authors have no data showing haplotype reconstruction or LD analysis to really show that it is high-resolution.

3. The detailed phenotyping of the NZM mouse and integration with the microbiome data appears disjointed to the rest of the paper. Is there a way to streamline or connect better?

Reviewer #1
1. (...) it does not appear that the study data have been deposited in a public repository, aside from the deposition of the genomic sequencing data. (...) Public deposition of the data would greatly increase the value of the study and likelihood of data re-use and further discovery. Recommend Dryad or Mouse Phenome Database.
Thank you very much for this important comment. In addition to WGS data, we have now deposited all the raw sequencing data i.e. FASTQ files for RNAseq, Microbiome and Mycobiome from the NZM2410/J mice at NCBI SRA with accession number "PRJNA543200". All scripts used in analysis of QTL mapping and processing of sequencing of NZM2410/J mice are publicly available at GitHub (https://github.com/YaskGupta/QTLxDiet-in-complex-Traits). In addition, all the raw genotype data from ALL mice cohort, processed VCF files from newly sequenced strains (NZM2410/J, BxD2/TyJ and MRL/MpJ) are available on the Dryad database (DOI 10.5061/dryad.c8gc64n) as suggested. Furthermore, the data can be visualized and explored at http://diet.ag-ludwig.com. Currently, the user ID for the database is "ralf" and password is "LiedLudwig!". Upon publication of our paper, the webserver will be accessible without any password restrictions. In the revised version of the manuscript, we have added the following sentences to the data deposition statement (page 22): "Whole genome sequencing data for the three founder strains has been submitted in public database European Nucleotide Archive (ENA). The accession numbers are ERS2905893 (NZM2410/J), ERS2905894 (MRL/MpJ) and ERS2905895 (BxD2/TyJ). All the raw sequencing data, i.e. FASTQ files for RNA-Seq, microbiome and mycobiome from NZM2410/J, was submitted to NCBI SRA with the accession number PRJNA543200. All scripts were also made publicly available at GitHub (https://github.com/YaskGupta/QTLxDiet-in-complex-Traits). In addition, all the genotype data is available on the Dryad database (DOI 10.5061/dryad.c8gc64n). The data can be visualized and explored at http://diet.ag-ludwig.com."
2. The opening paragraph of the introduction makes broad claims about heritability that seem to confound concepts of environmental and genetic effects. A more thoughtful discussion of how GxE could impact heritability estimation, with some up to date citations (e.g. work by Noah Zaitlen), would be helpful.
We agree with the reviewer that the manuscript indeed discusses gene environment interactions (GxE) rather than impact of GxE on heritability estimation. Therefore, we have changed the focus of our introduction towards GxE interactions. In this line, we also included citations from Noah Zaitlen. For review purposes only, we did calculate the "expected heritability" estimates for studied complex traits using LDAK software¹ (For-Review-Only Table 1).
3. But the work presented does not address the missing heritability problem. For example, there are no estimates of heritability or genetic variance reported for their study population. The few references provided are all from 2009 or earlier. Likewise the discussion of genetic of mouse populations is outdated and narrow in scope (two old references from the same group of researchers). In this vein, the title makes too bold of a claim and does not accurately reflect the content of the paper.
We thank reviewer 1 for these important comments. Regarding the missing heritability problem, please see our answer to your question #2. As we now focus more on GxE interactions rather than on the missing heritability, the respective paragraphs in the manuscript have been deleted. We have furthermore updated the reference list (e.g. from Noah Zaitlen), including newer citations, carefully re-read the manuscript, and changed the introduction and the title to better reflect the content of the paper.
4. The authors use the term "gender" when they actually mean "sex". Gender is a

social/cultural construct whereas sex refers to the biological state of an organism. Mice do have gender but the authors clearly mean sex.

As per your suggestion, throughout the manuscript the term „gender“ was replaced by “sex”.

Reviewer #2

1. Diet treatment: The treatment is generally described in the supplement material. Two aspects should be clarified:

a) 40% calorie restriction based on the ad libitum uptake of sex and aged-matched mice with higher body weight presumably does not affect energy balance too much especially at a later age stage. Mice likely compensate by reducing locomotor activity and metabolic rate at resting. Could that have an impact on the results?

b) I could not find detailed information on the Western diet. Regarding reproducibility I would recommend a detailed description of the experimental diets including details if the diet was a grain-based chow or a purified diet which would have considerable effects on the microbiome and likely also on metabolic functions. Regarding this context, I wonder if the authors were mainly interested in calorie consumption or specific diet composition (e.g. higher fat content). What was the experimental rationale?

Thank you for these important comments.

a) The reviewer raises an important point. We, however, did not monitor the mice for locomotor activity. We added this aspect as a limitation of our study in the discussion section in the revised version of the manuscript. Specifically we write: *“We did not monitor, however, the mice for their locomotor activity, which may have been impacted by the different diets⁶³⁻⁶⁵. Thus, this needs to be taken into consideration as a limitation when interpreting the data of our study.”*

b) We now added this information (new supplementary table 15) and provided a detailed overview of the composition of the purified diets used in the study. The overall experimental rationale was to mimic dietary lifestyles in their extremes, such as “normal” control diet, “western diet” mimicking the food of the modern western countries, as well as deficit of food intake in developing countries. We have also added the previous sentence on page 4 of the revised manuscript to make this more evident. In this line, we were not specifically focusing on either calorie intake or single dietary components. Focusing on the microbiome, we used purified diets to avoid significant changes of diet composition between different batches.

2. Personally, I still prefer the term “sex” as a main experimental factor in an animal study instead of “gender”.

Throughout the manuscript the term „gender“ was replaced by “sex”.

3. L 344 - LEfSe algorithm – I am not sure if every reader is familiar with that term – brief explanation?

Thanks! We have now added a brief explanation of the term. Specifically, we now write: *“(…) we used the LEfSe algorithm. The linear discriminant analysis effect size (LEfSe) method combines standard statistical tests with biological consistency and effect relevance to determine the features (taxonomical ranks) that most likely explain the differences between classes (such as diet and disease). For mycobium, at the transient stage, (…)”* on page 11 of the revised manuscript.

4. L 358 - Why “unceasingly” – is this term justified?

We agree with the reviewer and deleted the term “unceasingly”.

5. L 359 – “presence or absence of certain microbial species precedes the onset of the disease phenotype” – maybe a bit simplistic for complex pathophysiological changes over time.
We agree with the reviewer. We omitted these lines from the main text. In addition, we changed “disease onset” by “clinical disease manifestation” as lupus develops over time; and “disease onset”, as opposed to clinical disease manifestation”, is not precisely defined.
6. L 382 “FMCs” – very brief explanation may be helpful
We thank the reviewer for this helpful comment. We have added a brief explanation of the term to the main text, specifically (page 12 of the revised manuscript): “As previously shown, microbiota can be clustered into functional microbial communities (FMCs) based on taxa co-occurrences patterns. To determine such ecological structures, Tong and colleagues developed a methodology to infer microbial co-occurrence networks. Nodes of these networks, representing OTUs were grouped based on their topological overlaps using hierarchical clustering and were termed as FMCs.”
7. L 382 “eigenOTUs” see above – very brief explanation may be helpful
We thank the reviewer for this comment. We have added a brief explanation of the term to the main text. Same as above, please see page 12: “Such an approach provides dimensionality reduction (eigenOTUs; that can be described as first principle component of functional microbial communities to summarize the OTU abundances in a given community).”
8. In the discussion, I missed that the factor “sex” is not discussed in more detail.
We agree with the reviewer. We discussed the impact of sex on the complex traits in the revised version of the manuscript. Specifically, we write (on page 15): “Additionally, and in line with previous studies, when using sex as an interactive covariate, we identified predominantly QTL accounting for hematological parameters^{59, 60}. Recently, a systematic review of animal research showed a vast over-representation of experiments that exclusively included mice of a single sex in their experiments. Where two sexes were included, most of the data was analyzed without taking sex into account. Using sex as a biological variable, close to 10% of categorical traits and over 50% of continuous data exhibited sexual dimorphism⁶¹. Herein, we show a much lesser impact of sex on the variability of complex traits. This seeming discrepancy may be best explained by the difference in mouse phenotypes investigated.”
9. Overall, the data provide evidence for functional links but the study is still about associations and not causality. I think that needs to be stressed at some point in the discussion.
We thank reviewer 2 for this important comment. We have now modified the text and taken more caution not to mention causality when not supported by any evidence. Respective changes were made throughout the revised version of our manuscript.
10. Can the authors briefly discuss the translational value of their study?
We expanded the discussion towards the translational value of our study. Specifically we write (on page 17): “In terms of clinical translation, identifying gene-environment interaction may help to identify novel interventions that are beneficial for a defined subgroup of the population carrying a specific genotype⁷³. For instance, it is tempting to speculate, based on the results of our study, that lupus patients expressing lower levels of the TNXB gene are likely to benefit more from caloric restriction. Moreover, our results in the NZM2410/J mice indicate that dietary regulation of the microbiome is associated with lupus development. Thus, suggesting that dietary interventions and/or use of probiotics may be used as preventive measures in at risk populations.”

Reviewer #3

1. One major flaw in the general mouse genetic study with the AIL is that each mouse is unique and while you observe differences in phenotypes you have no way to know truly how the diet effects genetics because diet can not be compared across unique mice. While one could argue that the alleles are constantly recycling in the AIL, it would still make analysis of diet interactions quite difficult. How do the authors reconcile this fact? There are quite a few mouse genetics studies that show similar effects

We agree with the reviewer that addressing GxE interactions in an experimental setup is challenging. However, mimicking the impact of diet in humans as an example for an environmental factor on phenotypic variation can only be studied when different genotypes are present in the studied population. In order to create this required diversity, we generated the genetically diverse AIL mice by intercrossing for several generations. We also agree that GxE effects on complex traits are marginal and difficult to identify. To address this challenge, we used a large cohort of mice (1,154) to increase the power of our study and to detect true positive signals. Additionally, to eliminate any false positive associations of complex traits with GxE, we focus on only strong associations (genome-wide threshold < 0.05 derived from 1,000 permutations). Previously, such GxE associations have been observed not only in studies using mice^{2, 3, 4, 5, 6, 7}, but also in other organisms^{8, 9, 10, 11, 12}. Furthermore, our approach has identified several previously validated candidate genes such as *Napepld* to be associated with body weight. Previously, deletion of this gene led to obesity in mice¹³. To further address your concern, we have re-written the manuscript emphasizing the impact of GxE interactions on complex phenotypes, rather than on how diet affects genetic heritability.

2. I am concerned about that the high-resolution mapping is greatly overstated. The authors have no data showing haplotype reconstruction or LD analysis to really show that it is high-resolution.

Prompted by your comment, we performed LD analysis based on the genotyping data using PLINK¹⁴. Afterwards, we estimated the haplotype blocks in our cohort of mice using the sliding window size ranging from 0.5 up to 10 Mb. Note, the default sliding window size in PLINK software is 0.2 Mb which is used for human studies. We show that the average block size is around 0.35 Mb \pm 0.6 Mb even at 10 Mb window. Detailed LD plot across each chromosome and summary of haplotype block size estimation is provided within this review (For-Review-Only figures 1-2). However, to further address your concern, we have modified the title of our last paragraph to “*Fine-mapping QTL using whole genome sequencing of founder strains*”.

3. The detailed phenotyping of the NZM mouse and integration with the microbiome data appears disjointed to the rest of the paper. Is there a way to streamline or connect better?

The NZM data has been included because at the end of the paragraph “QTL fine-mapping using whole genome sequencing of founder strains”, we had demonstrated that diet considerably shifts the genetic association and uncovered multiple genes associated with metabolic and pathophysiological traits. To address, if this diet-mediated effect of genetic association is of functional relevance, we decided to investigate the impact of diet on lupus manifestation in the NZM2410/J mouse because (i) this strain was one of 2 major contributors to genetic variation and (ii) it develops lupus, which is associated with ANA, for which we had identified a QTL in the AIL population. The subsequently performed RNA-Seq and micro-/mycobiaota analysis was initiated to unravel potential mechanisms, as well as to demonstrate that multi-dimensional datasets can be used for further fine-mapping of genetic susceptibility. To further streamline the text on page 10 of the revised manuscript, we now write: “*Next, to delineate changes induced by diet that resulted in differential susceptibility to lupus, we studied gut flora (longitudinally) and transcriptomic alterations in these mice. (...). Therefore, we performed longitudinal sampling of feces from lupus-prone NZM2410/J mice that were set on the same diet as the AIL mice. We categorized samples into three stages i.e. 1) samples (...). Afterwards, using amplicon-*

based next generation sequencing, samples were investigated for microbial (V1-V2) and mycobial (ITS2) composition."

Legends for For-Review-only-figures:

Figure R1. Heatmap representing linkage disequilibrium (R^2) between adjacent markers in each of the chromosomes in the AIL mouse cohort.

Figure R2. Summary statistics for estimated haplotype blocks in the AIL mouse cohort. The figure shows windows size (0.5-10 Mb) in x-axis and estimated haplotype block size in y-axis. The statistics include mean, median, SD (standard deviation), GM (geometric mean), Q1 (first quartile), and Q2 (third quartile).

References

1. Speed D, Hemani G, Johnson MR, Balding DJ. Improved heritability estimation from genome-wide SNPs. *Am J Hum Genet* **91**, 1011-1021 (2012).
2. Engstrom AK, Snyder JM, Maeda N, Xia Z. Gene-environment interaction between lead and Apolipoprotein E4 causes cognitive behavior deficits in mice. *Mol Neurodegener* **12**, 14 (2017).
3. Lin C, *et al.* QTL analysis of dietary obesity in C57BL/6byj X 129P3/J F2 mice: diet- and sex-dependent effects. *PLoS One* **8**, e68776 (2013).
4. Cheng Y, *et al.* Body composition and gene expression QTL mapping in mice reveals imprinting and interaction effects. *BMC Genet* **14**, 103 (2013).
5. Leamy LJ, Kelly SA, Hua K, Pomp D. Exercise and diet affect quantitative trait loci for body weight and composition traits in an advanced intercross population of mice. *Physiol Genomics* **44**, 1141-1153 (2012).
6. Leamy LJ, Gordon RR, Pomp D. Sex-, diet-, and cancer-dependent epistatic effects on complex traits in mice. *Front Genet* **2**, 71 (2011).
7. Tyler AL, *et al.* Epistatic Networks Jointly Influence Phenotypes Related to Metabolic Disease and Gene Expression in Diversity Outbred Mice. *Genetics* **206**, 621-639 (2017).
8. Dahl A, *et al.* Reverse GWAS: Using genetics to identify and model phenotypic subtypes. *PLoS Genet* **15**, e1008009 (2019).

9. Unckless RL, Rottschaefer SM, Lazzaro BP. The complex contributions of genetics and nutrition to immunity in *Drosophila melanogaster*. *PLoS Genet* **11**, e1005030 (2015).
10. Winham SJ, Biernacka JM. Gene-environment interactions in genome-wide association studies: current approaches and new directions. *J Child Psychol Psychiatry* **54**, 1120-1134 (2013).
11. Cornelis MC, *et al.* Gene-environment interactions in genome-wide association studies: a comparative study of tests applied to empirical studies of type 2 diabetes. *Am J Epidemiol* **175**, 191-202 (2012).
12. Rask-Andersen M, Karlsson T, Ek WE, Johansson A. Gene-environment interaction study for BMI reveals interactions between genetic factors and physical activity, alcohol consumption and socioeconomic status. *PLoS Genet* **13**, e1006977 (2017).
13. Geurts L, *et al.* Adipose tissue NAPE-PLD controls fat mass development by altering the browning process and gut microbiota. *Nat Commun* **6**, 6495 (2015).
14. Purcell S, *et al.* PLINK: a tool set for whole-genome association and population-based linkage analyses. *Am J Hum Genet* **81**, 559-575 (2007).

REVIEWERS' COMMENTS:

Reviewer #1 (Remarks to the Author):

I am happy with the changes that the authors made in response to the review comments.

Reviewer #2 (Remarks to the Author):

In my view all issues raised were sufficiently addressed.

Reviewer #3 (Remarks to the Author):

..

Reviewer #1: I am happy with the changes that the authors made in response to the review comments.

Thank you very much!

Reviewer #2: In my view all issues raised were sufficiently addressed.

Thank you very much!

Reviewer #3: ----

Thank you very much!